

# Permafrost temperature baseline at 15 meters depth in the Qinghai-Tibet Plateau (2010–2019)

Defu Zou[1], Lin Zhao[2*], Guojie Hu[1], Erji Du[1], Guangyue Liu[1], Chong Wang[2], Wangping Li[3]

[1]Cryosphere Research Station on Qinghai–Tibet Plateau, Key Laboratory of Cryospheric Science and Frozen Soil Engineering,
Northwest Institute of Eco–Environment and Resources (NIEER), Chinese Academy of Sciences (CAS), Lanzhou, 730000, China
[2]School of Geographical Sciences, Nanjing University of Information Science & Technology, Nanjing, 210044, China
[3]School of Civil Engineering, Lanzhou University of Technology, Lanzhou, 730050, China

*Correspondence to*: Lin Zhao (lzhao@nuist.edu.cn)

**Abstract.** The ground temperature at a fixed depth is a crucial boundary condition for understanding the properties of deep permafrost. However, the commonly used mean annual ground temperature at the depth of the zero annual amplitude ($MAGT_{dzaa}$) has application limitations due to large spatial heterogeneity in observed depths. In this study, we utilized 231 borehole records of mean annual ground temperature at a depth of 15 meters ($MAGT_{15m}$) from 2010 to 2019 and employed
support vector regression (SVR) to predict gridded $MAGT_{15m}$ data at a spatial resolution of nearly 1 km across the Qinghai-Tibet Plateau (QTP). SVR predictions demonstrated a $R^2$ value of 0.48 with a negligible negative overestimation (-0.01 °C). The average $MAGT_{15m}$ of the QTP permafrost was -1.85 °C (±1.58 °C), with 90% of values ranging from -5.1 °C to -0.1 °C and 51.2% exceeding -1.5 °C. The freezing degree days (FDD) was the most significant predictor (p<0.001) of $MAGT_{15m}$, followed by thawing degree days (TDD), mean annual precipitation (MAP), and soil bulk density (BD) (p<0.01). Overall, the
$MAGT_{15m}$ increased from northwest to southeast and decreased with elevation. Lower $MAGT_{15m}$ values are prevail in high mountainous areas with steep slopes. The $MAGT_{15m}$ was the lowest in the basins of the Amu Darya, Indus, and Tarim rivers (-2.7 to -2.9 °C) and the highest in the Yangtze and Yellow River basins (-0.8 to -0.9 °C). The baseline dataset of $MAGT_{15m}$ during 2010-2019 for the QTP permafrost will facilitates simulations of deep permafrost characteristics and provides fundamental data for permafrost model validation and improvement.

**Keywords:** Mean annual ground temperature; Permafrost; Support vector regression; Qinghai-Tibet Plateau

## 1 Introduction

The ground temperature at a given depth is a fundamental indicator for characterizing the thermal state of permafrost (Romanovsky et al., 2010). However, obtaining it at a large depth is challenging due to the harsh climatic conditions in permafrost regions and time-consuming drilling operations (Zhao et al., 2024). Previous studies have commonly used the mean
annual ground temperature (MAGT) at the depth of the zero annual amplitude (DZAA, representing the maximum depth that





seasonal surface temperature variations can penetrate) (Dobinski, 2011). Biskaborn et al. (2019) used records of the MAGT at the DZAA ($MAGT_{dzaa}$), demonstrating that most of the permafrost had experienced warming at a global scale with various magnitudes. The rate of increase in the $MAGT_{dzaa}$ is approximately 1 °C per decade in colder permafrost regions in the high-latitude Arctic and 0.3 °C per decade in warmer permafrost in the sub-Arctic regions (Smith et al., 2022).

In addition to indicating permafrost warming at a specific location, the assembled $MAGT_{dzaa}$ records can also be utilized to map regional permafrost occurrence using spatialization methods. For instance, Aalto et al. (2018) produced a map of circum-Arctic permafrost based on $MAGT_{dzaa}$ derived from statistical forecasting models. However, these maps often do not include permafrost extent to the south of 30°N in the Northern Hemisphere. Although the European Space Agency (ESA) Climate Change Initiative (CCI) provides permafrost MAGT products for the Northern Hemisphere (Obu et al., 2021), the deepest
depth reached is only 10 meters, limiting the applicability to regions where the DZAA exceeds 10 meters, such as the Qinghai-Tibet Plateau (QTP). Recently, Ran et al. (2022) updated the Northern Hemisphere permafrost map by incorporating more observed $MAGT_{dzaa}$ data from the QTP and northeast China and employing multiple machine-learning models, demonstrating significant advancements in mapping permafrost distribution and thermal state.

A critical consideration is the DZAA variability across different regions because it depends on permafrost dynamics.
Measurements from 1002 boreholes have shown that the DZAAs ranged from approximately 3 to 25 meters in different permafrost regions (Ran et al., 2022). The DZAAs are typically shallower in peat and mineral soils and deeper in bedrock (Smith et al., 2010). The DZAAs generally range from 10 to 15 meters in central Asia, depending on surface land cover and substrate properties (Zhao et al., 2010). Furthermore, DZAAs undergo change with permafrost warming, as evidenced by in situ observations indicating a decrease rate of 0.14-0.17 meters per year near the northern permafrost limit on the QTP from
2005 to 2017, leading to consequent changes in $MAGT_{dzaa}$ (Liu et al., 2021). The spatial and temporal variability of DZAAs complicates the use of the predicted $MAGT_{dzaa}$ maps for comparison and calibration with transient modeling results at specific depths and also limits their utility in estimating deeper permafrost characteristics (e.g., permafrost thickness). Hence, establishing a baseline of MAGT at a specific depth and for a specific period is crucial for permafrost modeling studies.

The QTP is the largest permafrost region in low and mid-latitudes and a typical high-altitude permafrost area (Zou et al., 2017).
Over the past two decades, permafrost monitoring efforts on the QTP have established a substantial monitoring network and some datasets have been published (Zhao et al., 2021). Particularly since 2010, extensive monitoring has been conducted by various research groups in multiple regions, resulting in updated datasets. This study aims to establish a fixed-depth deep permafrost temperature baseline using data from the QTP for the last decade (2010-2019) and a machine learning approach to address the limitations associated with the use of $MAGT_{dzaa}$. MAGT data at 15 meters depth are used for spatialization,
considering that DZAAs generally ranges from 10 to 15 meters in the QTP (Zhao et al., 2010). The resulting dataset can serve as a reference for model simulations during this period and provide an upper boundary condition for estimating deeper permafrost characteristics.



## 2 Materials and Methods

### 2.1 Compilation and Processing of MAGT Data

The dataset of mean annual ground temperature at 15 m depth ($MAGT_{15m}$) across the QTP permafrost region consisted of a total of 231 boreholes (Fig. 1). For this dataset, 122 measurements were obtained from the permafrost monitoring networks of the QTP (Zhao et al., 2021) established by the Cryosphere Research Station, Chinese Academy of Sciences (CRS-CAS). The remaining 109 measurements were compiled from published articles (Cao et al., 2018; Li et al., 2012, 2014, 2016; Luo et al., 2012, 2013, 2018; Sun et al., 2018; Liu and Shi, 2019). The observational period was from 2010 to 2019. The ground

temperature in the boreholes was measured using a cable equipped with a string of thermistors at various depths. The thermistor probe was developed and calibrated by the State Key Laboratory of Frozen Soil Engineering (SKLFSE, CAS), ensuring a measurement accuracy of ±0.05 °C in laboratory conditions (Wu et al., 2010).

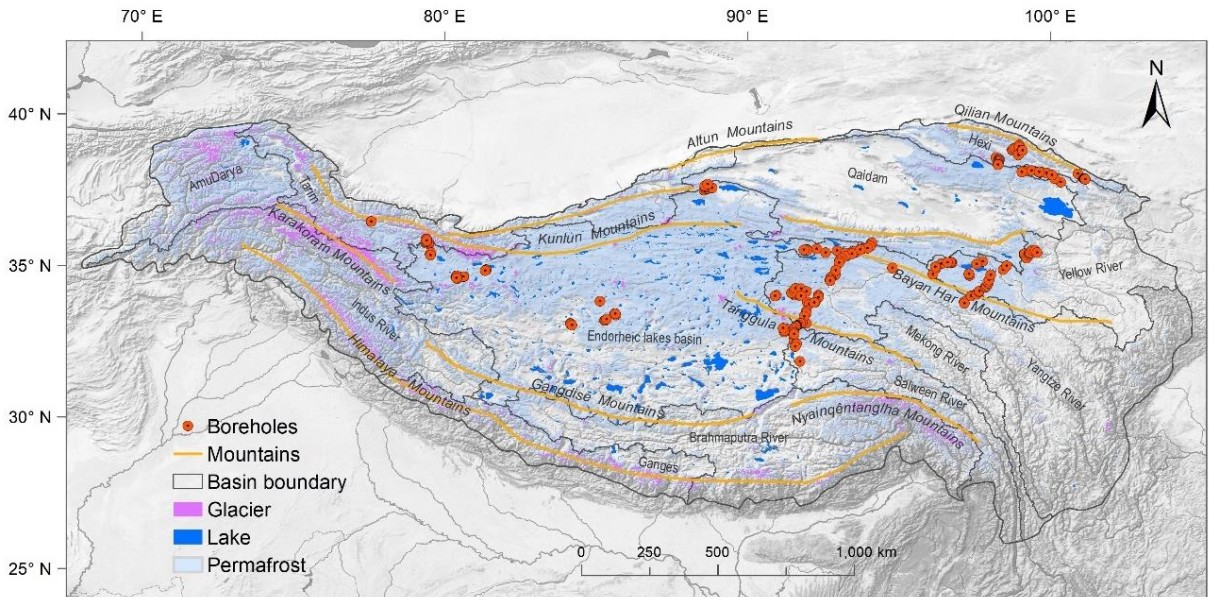

**Figure 1: Distribution of boreholes (n=231) for monitoring mean annual ground temperature at 15 m depth ($MAGT_{15m}$).**

Of the 231 boreholes, 180 sites (approximately 78%) only had single-year observations. To ensure that each site had values for each year from 2010 to 2019, we implemented the following processes:

1) For sites with multiple years of $MAGT_{15m}$ observations, we calculated the warming rates from 2010 to 2019 (Fig. 2a) and established the linear relationship between the warming rates and the average $MAGT_{15m}$ (2010-2019) (Fig. 2b). A rigorous selection process was used for sites with a minimum requirement of three observation years and a time span of six years. This

selection criterion ensured that the chosen sites provided a robust basis for calculating the warming rate. Consequently, 51



sites were included in the analysis. The $MAGT_{15m}$ range for these sites was -3.95 °C to 0.03 °C, encompassing a diverse temperature range in the dataset.

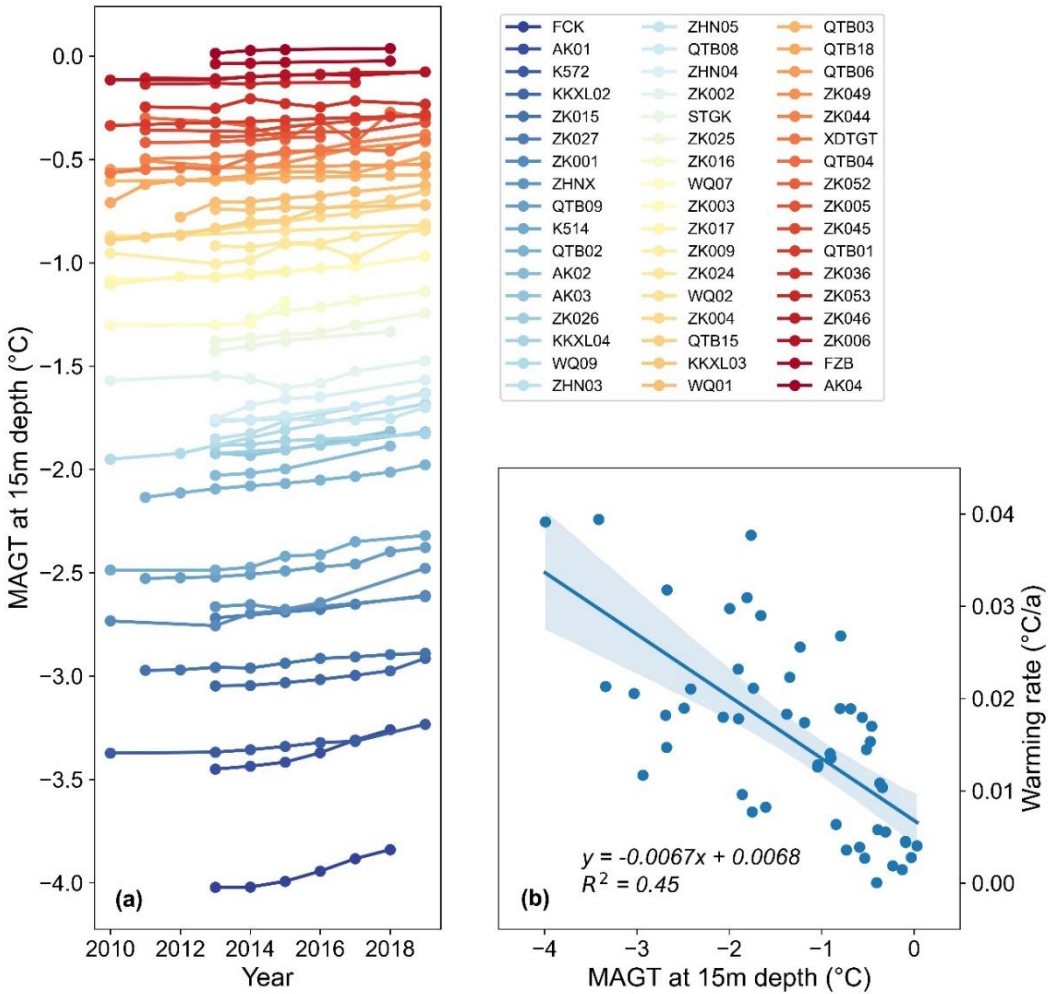

**Figure 2: Warming rates of MAGT$_{15m}$ during 2010-2019 (a) and the relationship between warming rates and the average MAGT$_{15m}$ (b).**

2) For sites that did not meet the first criterion, i.e., sites with observation years of less than three years or a time span of less than six years (a total of 180 sites), we used the following linear relationship (Equation 1) to fill in the missing values.

$$MAGT_{15m}\ warming\ rate = -0.0067 \times MAGT_{15m} + 0.0068 \quad (R^2 = 0.4549) \tag{1}$$

where, the *MAGT$_{15m}$ warming rate* represents the rate of warming of MAGT$_{15m}$ from 2010 to 2019, and *MAGT$_{15m}$* is the mean MAGT value at a depth of 15 m from 2010 to 2019.



After obtaining the yearly $MAGT_{15m}$ values for all sites from 2010 to 2019, we computed the mean $MAGT_{15m}$ value during the ten years. It was used as the input for regional $MAGT_{15m}$ predictions. This approach ensured the integration of long-term trends and provided a representative estimate of the regional MAGT.

## 2.2 Statistical Learning Model

In this study, we employed support vector regression (SVR) method (Basak et al., 2007) to predict the $MAGT_{15m}$. Although several machine learning models, including generalized linear model (GLM), generalized additive model (GAM), random forest (RF), and geographically weighted regression (GWR), have been commonly used in various regression analyses (Aalto et al., 2018; Wang et al., 2020; Ran et al., 2022), the SVR method has demonstrated better performance for predicting the MAGT in the QTP permafrost region compared to other methods (Ran et al., 2021). Moreover, SVR is a deterministic

prediction method, ensuring consistent and reproducible results for a fixed set of sample points, contributing to the reliability and replicability of predictions.

The SVR was implemented using the R package e1071. SVR is a nonparametric technique that seeks to find a function deviating from observations by a value not exceeding a threshold ($\varepsilon$) for the training points while minimizing the prediction error. The output model depends on kernel functions, the default radial kernel function was utilized in this study. Model

parameter selection was performed using a tuning method, with a cost parameter of 1000 employed to prevent overfitting and a gamma value of 0.0001. Model performance was assessed using the bias, root-mean-square error (RMSE), and coefficient of determination ($R^2$) computed via 10-fold cross-validation with 1000 repetitions. In each run, 90% of the measurements were used to train the SVR model, while the remaining 10% were used to test predictions.

## 2.3 Environmental and Climate Data

Nine environmental and climate variables were selected as predictors in the prediction of $MAGT_{15m}$ using the SVR method. They were selected based on previous studies (Aalto et al., 2018; Ran et al., 2021), and these variables were derived from high-quality datasets available at present (see Table 1).

Volumetric coarse fragments (CF, %) and bulk density (BD, g/cm$^3$) in the soils were obtained from the SoilGrids 2.0 data (Poggio et al., 2021) with a spatial resolution of 1 km by 1 km. In addition to soil texture, soil organic carbon content (SOC,

g/kg) data from the Third Pole (Wang et al., 2021) was also used. Average values for a depth range of 0-2 m were used for all soil factors as model inputs. The number of freezing degree days (FDD, °C-days) and thawing degree days (TDD, °C-days) were calculated following the method of Zou et al. (2017), and the data period was expanded to 2003-2019. Mean annual precipitation (MAP, mm) from 1970 to 2000 was derived from the WorldClim version 2.1 dataset (Fick and Hijmans, 2017). Snow cover duration (SCD, days) from 2003 to 2020 was derived from the MODIS daily cloud-free snow cover product (Qiu

et al., 2021). The multi-year averages of FDD, TDD, MAP, and SCD were calculated and used as model inputs. The elevation



was obtained from a dataset compiled by Amatulli et al. (2018). The multi-year averaged maximum value of the normalized difference vegetation index (NDVI) from 2000 to 2021 was derived from the MOD13A2 products (Wang et al., 2022).

**Table 1. Environmental predictors used in the modeling.**

| Predictor | Unit | Data Source |
|---|---|---|
| Coarse fragments (CF) | % | Poggio et al., 2021 |
| Bulk density (BD) | g/cm$^3$ | (SoilGrids 2.0) |
| Soil organic carbon content (SOC) | g/kg | Wang et al., 2021 |
| Freezing degree days (FDD) | °C-days, average for 2003–2019 | expanded the data of Zou et al., 2017 |
| Thawing degree days (TDD) | °C-days, average for 2003–2019 | |
| Mean annual precipitation (MAP) | mm, average for 1970-2000 | Fick and Hijmans, 2017 (WorldClim version 2.1) |
| Snow Cover Duration (SCD) | days, average for 2003-2020 | Qiu et al., 2021 |
| Elevation (DEM) | m | Amatulli et al., 2018 |
| Normalized difference vegetation index (NDVI) | maximum value in a year, average for 2000-2021 | Wang et al., 2022 (MOD13A2 products) |

**2.4 Ancillary Data**

Prior to calculating statistics, the SVR predictions were resampled to a spatial resolution of 1 km by 1 km. Additionally, areas covered by glaciers and lakes were masked from the output results. Glacier areas were masked using the Randolph Glacier Inventory (RGI6.0) data obtained from the National Snow and Ice Data Center (https://nsidc.org/data/nsidc-0770/versions/6). Lake areas were masked by referencing the dataset of "the lakes larger than 1 km$^2$ in the Tibetan Plateau (v3.1) (1970-2022)" (Zhang et al., 2019) provided by the National Tibetan Plateau Data Center (http://data.tpdc.ac.cn/).

**3 Results**

**3.1 Model Performance**

The cross-validation of 1000 runs demonstrated that the SVR model achieved high accuracy. The mean values of the three statistical indicators, i.e., bias, root-mean-square error (RMSE), and coefficient of determination ($R^2$) were -0.01 °C (±0.11 °C), 0.71 °C (±0.13 °C), and 0.48 (±0.14), respectively. Figure 3 shows the scatterplot depicting the relationship between the predicted MAGT$_{15m}$ values and the observed measurements. The fitted line is close to the 1:1 line, with a bias and RMSE of 0.01 °C and 0.73 °C, respectively, indicating a close agreement with the cross-validation results. The predictions exhibited slight underestimations at high MAGT$_{15m}$ values, while overestimations at lower ground temperatures.

Multilinear regression analysis revealed that the contribution of FDD to the MAGT$_{15m}$ prediction was highly significant ($p<0.001$), whereas those of the TDD, MAP, and BD were significant ($p<0.01$). However, the contributions of the remaining five factors (DEM, SCD, SOC, CF, and NDVI) were insignificant. Overall, at the plateau scale, the ground surface temperature

(especially for FDD), precipitation, and soil bulk density contributed the most to $MAGT_{15m}$, whereas the other environmental and climate factors modified this influence at the regional scale.

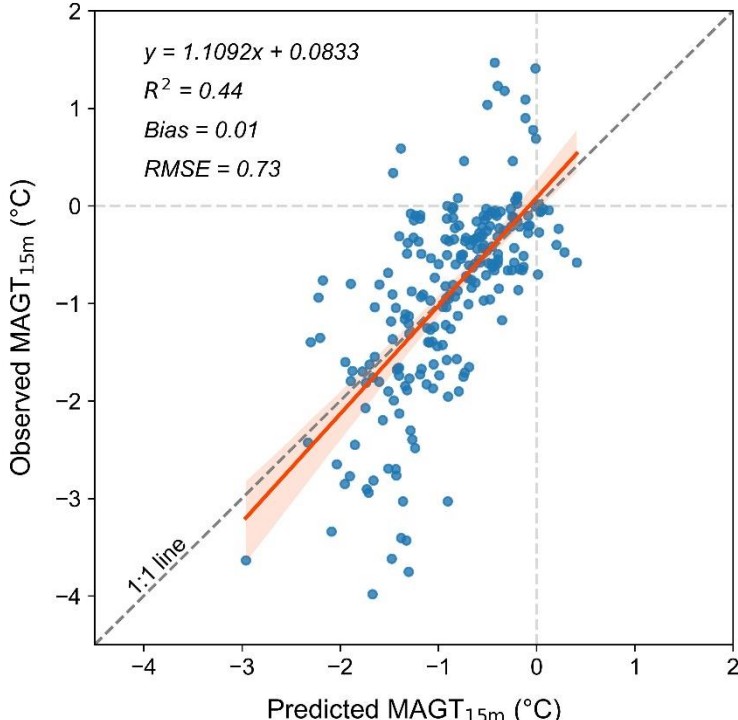

**Figure 3: Relationship between predicted and observed mean annual ground temperatures at 15 m depth ($MAGT_{15m}$).**

145 **3.2 Distribution Characteristics**

**3.2.1 General distribution characteristics of $MAGT_{15m}$**

Figure 4 illustrates the spatial distribution pattern of the predicted $MAGT_{15m}$ in the QTP for the period 2010-2019. Overall, the $MAGT_{15m}$ exhibited an increasing trend from northwest to southeast spatially. Regions with the lowest $MAGT_{15m}$ values (depicted as dark blue in Fig. 4) were primarily located in the high mountain regions of the western QTP, e.g., the western

150 Kunlun, Karakoram, and western Himalaya mountains (as displayed in Fig. 1). Regions with lower $MAGT_{15m}$ values were predominantly found in the central Kunlun, Tanggula, and Qilian mountains of the northern QTP. The $MAGT_{15m}$ increased south- and eastward, with high values concentrated in the majority of the Yangtze and Yellow River source areas, as well as the southern areas of the Endorheic Lakes basin. In other high mountainous regions, such as the Altun, Gangdise, eastern Himalayas, and Nyainqêntanglha mountains, the $MAGT_{15m}$ changed rapidly from high to low values over short distances due

155 to the steep terrain.

The region with negative MAGT$_{15m}$ values covered an area of 1.36×10$^6$ km$^2$ (excluding glacier and lake areas), accounting for approximately 44.0% of the total QTP area. The average MAGT$_{15m}$ was -1.85 °C with a standard deviation of ±1.58 °C, and 90% of the values ranged from -5.1°C to -0.1°C. The zoning statistics of the temperature indicated that the area with extremely low values was relatively small. The area with MAGT$_{15m}$ values below -5 °C was 0.07×10$^6$ km$^2$, accounting for only 5.1% of the region with negative MAGT$_{15m}$ values. The areas with MAGT$_{15m}$ values below -3 °C was 0.25×10$^6$ km$^2$, representing 18.1% of the region with negative MAGT$_{15m}$ values. The areas for the temperature ranges of -3 to -2 °C, -2 to -1 °C, and -1 to 0 °C were 0.24, 0.38, and 0.49×10$^6$ km$^2$, accounting for 17.8%, 28.0%, and 36.1%, respectively.

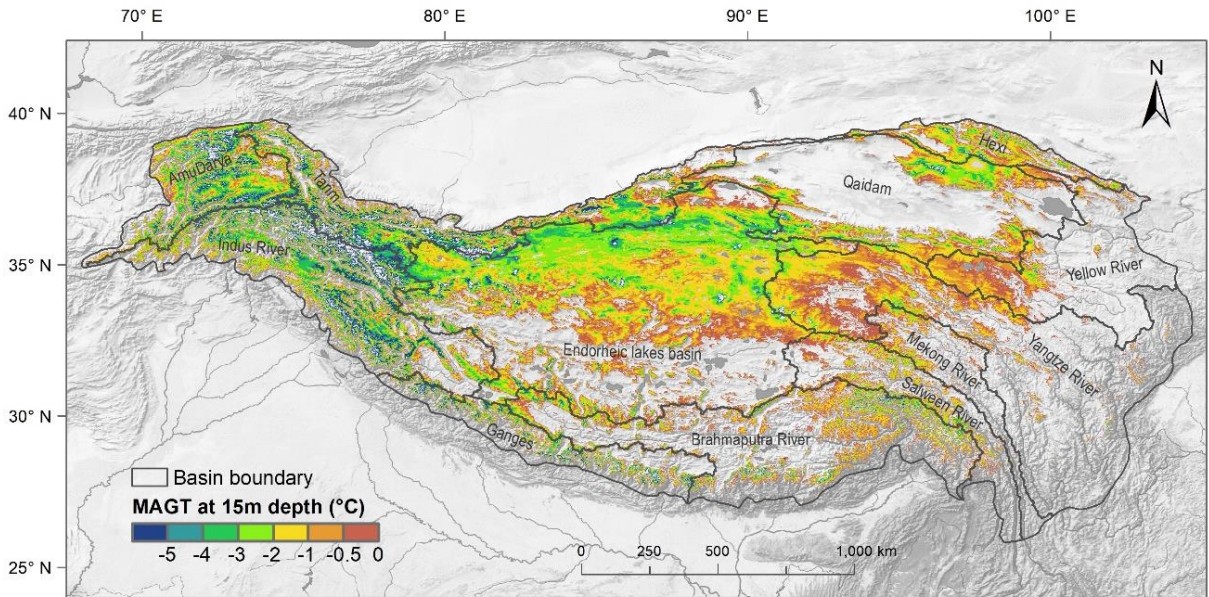

**Figure 4: Spatial distribution of predicted mean annual ground temperatures at 15m depth (MAGT$_{15m}$) across the Qinghai-Tibet Plateau.**

The three-dimensional ground thermal states across the QTP were investigated based on the predicted MAGT$_{15m}$. Figure 5a illustrates the distribution of the MAGT$_{15m}$ at different elevations and the percentage of area. Area analysis reveals that approximately 90% of the MAGT$_{15m}$ values fell in the elevation range of 3840 to 5570 m. Generally, the MAGT$_{15m}$ demonstrated a decreasing trend as elevation increased, and the variability in MAGT$_{15m}$ values was more pronounced at higher elevations and less significant at lower elevations. Meanwhile, the lapse rate of MAGT$_{15m}$ was relatively low at lower elevations and increased as the elevation rises.

The MAGT$_{15m}$ in the west-to-east longitudinal transect increased monotonically (Fig. 5b). The western regions of the plateau exhibited the lowest MAGT$_{15m}$ values (e.g., approximately -3 °C in the longitude range of 72°E to 78°E), whereas the easternmost areas had the highest values (approximately -1°C to the east of 100°E). Concurrently, the amplitude of the temperature variations gradually declined as the MAGT$_{15m}$ values increased from west to east. Figure 5c illustrates the

MAGT$_{15m}$ trend in the latitudinal transect. The regional average MAGT$_{15m}$ was generally stable at approximately -1.5 °C from 28°N in the Himalayas to 33°N in the Tanggula Mountains with a stable standard deviation. The MAGT$_{15m}$ decreased slightly from 33°N northwards, and the average was lower than in areas south of 33°N. The lowest latitudinal MAGT$_{15m}$ values occurred near 36°N, primarily in the Kunlun and Karakoram Mountain regions.

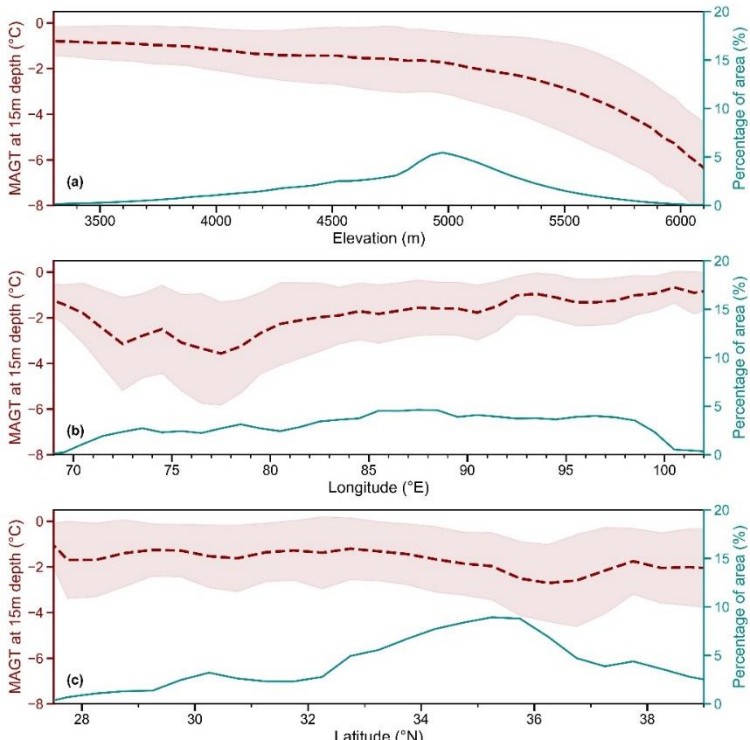

180

**Figure 5: Mean annual ground temperature at 15m depth (MAGT$_{15m}$) along elevation (a), longitude (b), and latitude (c) transects in the Qinghai-Tibet Plateau (the dashed red line represents the mean MAGT$_{15m}$, and the light-red shaded area indicates its standard deviation; the cyan dashed line shows the percentage of area).**

Due to the predominant mountainous and high plateau terrain of the QTP, we utilized slope as a topographical indicator to assess the spatial distribution characteristics of the MAGT$_{15m}$. Considering the slope distribution pattern within the study areas, we aggregated the slope gradients into four classes: flat (slope < 2°), gentle (2 to 8°), moderate (8 to 17°), and steep (> 17°). Statistical analysis reveals that the percentage of area (PA, %) of flat, gentle, and moderate slopes was 16.2%, 25.8%, and 20.5%, respectively. Regions with steep slopes comprised a significant portion, with a PA of 37.5% (Fig. 6a). Overall, the average MAGT$_{15m}$ exhibited a decreasing trend as the slope increased, indicating that lower MAGT$_{15m}$ values were more prevalent in areas with steeper slopes (e.g., high mountainous regions of the QTP).

Additionally, we analyzed the distribution of the four slope classes in three MAGT$_{15m}$ intervals: low (< -3 °C), medium (-3 to -1.5 °C), and high (> -1.5 °C) (Fig. 6b). The areas of low-, medium-, and high-temperature regions were 24.5, 41.6, and





$69.5 \times 10^4$ km$^2$, accounted for 18.1%, 30.7%, and 51.2%, respectively. Most of the MAGT$_{15m}$ values in the low-temperature interval were concentrated in regions with steep slopes and accounting for 60.6% of the total area. The areas of moderate slopes account for 21.4%, and the combined areas of flat and gentle slopes represented only 18.0% of the total. The areas of flat, gentle, moderate, and steep slopes accounted for 17.9%, 29.5%, 19.5%, and 33.1% in the medium-temperature interval and for 19.6%, 27.7%, 20.7%, and 32.0% in high-temperature interval, respectively, indicating similar proportions. Moreover, the areas of flat and gentle slopes were predominant in the medium- and high-temperature intervals, representing more than half of the total area in each interval. Topographically, MAGT$_{15m}$ values in high- and medium-temperature intervals (i.e., MAGT$_{15m}$ values exceeding -3 °C) occurred substantially in areas with flat and gentle slopes, whereas low-temperature regions were rare on these slopes.

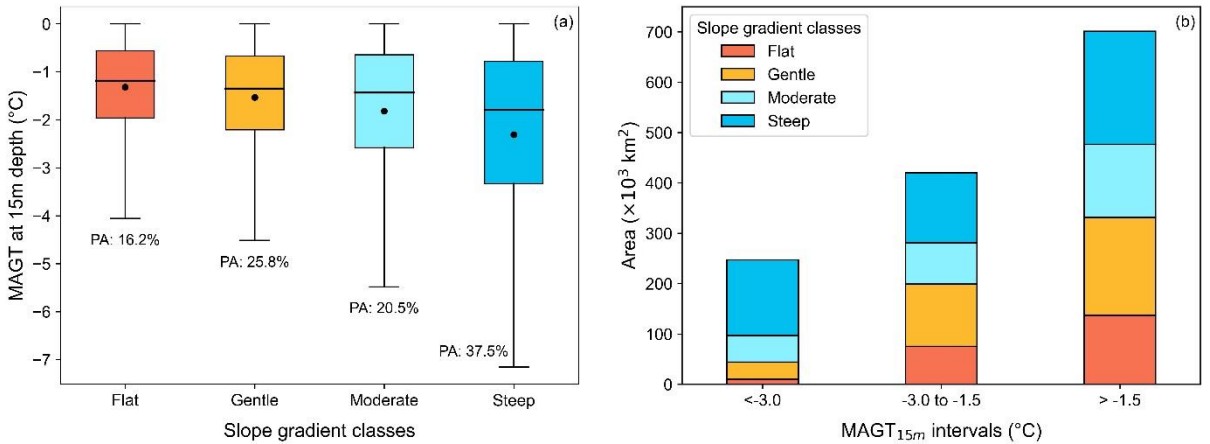

**Figure 6: Mean annual ground temperature at 15 m depth (MAGT$_{15m}$) for different slope gradients (PA: percentage of area).**

### 3.2.2 Regional distribution characteristics of MAGT$_{15m}$

To explore the spatial variability in different regions, we conducted the zonal statistics of the MAGT$_{15m}$ for 12 primary river or lake basins in the QTP (Fig. 7). Figure 7a and 7b show the box plots and cumulative plots of the three temperature intervals (<-3 °C, -3 to -1.5 °C, and >-1.5 °C) of the MAGT$_{15m}$ values in the basins, respectively. Figure 7 is organized primarily from west to east based on the location of the basins, showing an overall increasing trend in the MAGT$_{15m}$, consistent with the longitudinal profile results (Fig. 5b).

The lowest MAGT$_{15m}$ values were observed in the basins of the Amu Darya, Indus, and Tarim rivers in the western QTP, with average MAGT$_{15m}$ values of -2.69 °C, -2.93 °C, and -2.80 °C, respectively. These three basins also exhibited the largest standard deviations, ranging from ±1.89 to ±2.11 °C. Low MAGT$_{15m}$ values occurred throughout these basins, with areas of MAGT$_{15m}$ < -3°C comprising 37.7% to 42.4% and those < -5°C comprising 12.0% to 17.0% of the total basin area. Another region with low MAGT$_{15m}$ values and high variations was the Ganges River basin in the southernmost QTP. The mean



MAGT$_{15m}$ was -2.30 °C (±1.84 °C), and areas of MAGT$_{15m}$ < -3°C and < -5°C comprised 31.4% and 10.1% of the total basin area, respectively.

The average MAGT$_{15m}$ value in the Endorheic Lakes' basin was -1.73 °C, higher than in the western basins, with a small standard deviation of ±1.17 °C. This result indicates minimal spatial variation in the MAGT$_{15m}$ values, despite the basin being the most extensive permafrost area of the QTP. Few low MAGT$_{15m}$ values were observed in this basin, accounting for only
12.1% and 1.4% of areas with MAGT$_{15m}$ < -3°C and < -5°C, respectively. The Brahmaputra River basin exhibited the highest mean MAGT$_{15m}$ among the western and central QTP basins, reaching -1.31 °C with a small deviation of ±1.21 °C.

The six river basins in the eastern QTP generally exhibited higher MAGT$_{15m}$ values with smaller fluctuations. The Qaidam and Hexi basins in the northeastern QTP had lower MAGT$_{15m}$ values of -1.59 °C (±1.19 °C) and -1.44 °C (±1.11 °C), respectively. In the southeastern QTP, the Salween and Mekong River basins exhibited MAGT$_{15m}$ values of -1.09 °C (±0.93 °C)
and -0.91 °C (±0.80 °C), respectively. These basins had few areas with MAGT$_{15m}$ values < -3 °C, with only 0.3% and 0.2% of the area falling below this threshold. Conversely, a substantial proportion had MAGT$_{15m}$ values > -1.5 °C, at 73.2% and 80.3%, respectively. The MAGT$_{15m}$ values were slightly higher eastward in the Yangtze and Yellow River basins, with averages of -0.92 °C (±0.74 °C) and -0.80 °C (±0.61 °C), respectively. The standard deviations in these basins were below 1 °C. Less than 0.2% of the area had MAGT$_{15m}$ values < -3 °C, whereas the majority had MAGT$_{15m}$ values > -1.5 °C, accounting for 81.8%
and 87.9%, respectively.

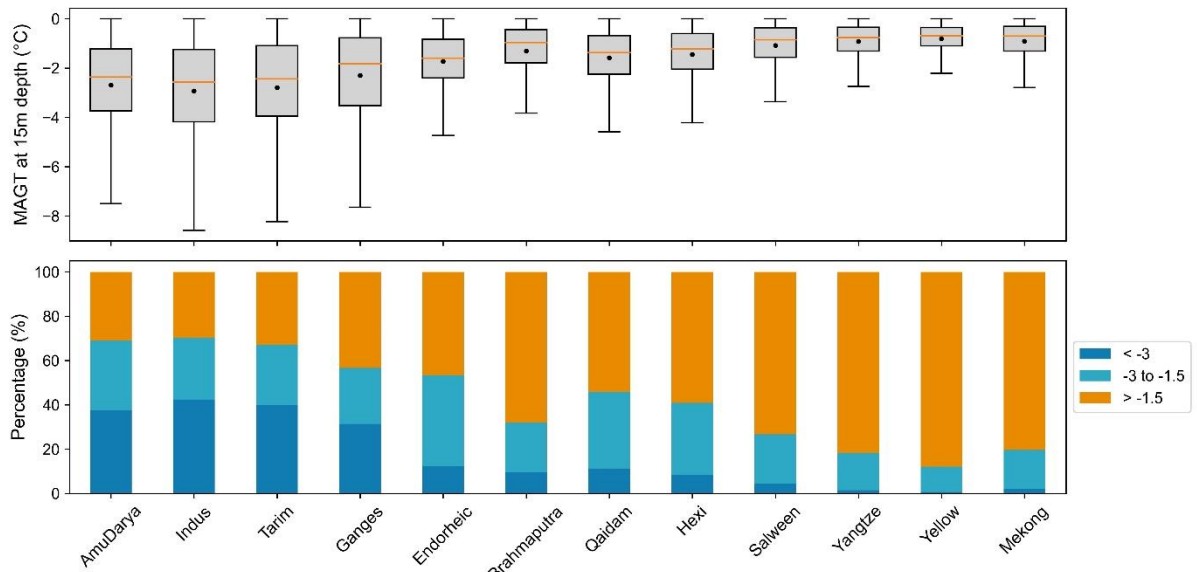

**Figure 7: Distribution (a) and percentage of area in three intervals (b) of MAGT at 15 m depth (MAGT$_{15m}$) in 12 basins of the Qinghai-Tibet Plateau.**



## 4 Discussion

The SVR method exhibited more scattered predictions in the low-temperature range compared to the high-temperature range, as illustrated in Figure 3. While performing well in the high-temperature range with slight overestimations, the SVR method may potentially result in some degree of underestimation in regions with lower temperatures. This variability could be attributed to the scarcity of $MAGT_{15m}$ observations with colder temperature. Notably, most the $MAGT_{15m}$ observations utilized in this study were obtained from the eastern QTP regions, particularly the Yangtze and Yellow River regions, which are

characterized by relatively higher $MAGT_{15m}$ values. Additionally, the analysis suggests that the vertical lapse rate of the $MAGT_{15m}$ increased with the elevation. This phenomenon indicates difference in the vertical lapse rate of the $MAGT_{15m}$ between higher and lower elevations. Hence, future monitoring efforts should prioritize cold permafrost areas to ensure adequate representation in the study area.

We considered negative $MAGT_{15m}$ values as indicators of permafrost and estimated the permafrost area as approximately

$1.36 \times 10^6$ km$^2$ (excluding glacier and lake areas), accounting for about 44.0% of the total QTP. This finding broadly aligns with previous studies based on $MAGT_{dzaa}$ reporting permafrost extents of approximately $1.30 \times 10^6$ km$^2$ by Wang et al. (2020) and $1.32 \times 10^6$ km$^2$ by Zhao et al. (2024). The discrepancies may arise from differences in data samples. Our study benefited from integrating $MAGT_{15m}$ records from 231 boreholes, providing a comprehensive dataset in terms of spatial coverage and quantity, ensuring the reliability of our prediction. Furthermore, the estimated permafrost area within the Chinese territory of

the QTP was approximately $1.11 \times 10^6$ km$^2$, comparable to values derived from the temperature at the top of the permafrost (TTOP) model (1.06 to $1.09 \times 10^6$ km$^2$) (Zou et al., 2017; Cao et al., 2023). Differences in measurement depths and temperature values between $MAGT_{dzaa}$ and $MAGT_{15m}$ may have contributed to slight variations in the permafrost area.

Based on the classification criteria established by $MAGT_{dzaa}$, the permafrost can be categorized into three types: cold ($\leq$ -3.0 °C), cool (-3 to -1.5 °C), and warm (> -1.5 °C) permafrost (Ran et al., 2022). We analyzed the distribution characteristics

of the QTP permafrost based on this classification system using the predicted $MAGT_{15m}$ data. Warm permafrost was the most prevalent type in the QTP, encompassing 51.2% of the permafrost region, followed by cool permafrost (30.7%) and cold permafrost (18.1%). The forms of permafrost degradation vary depending on ground temperature regimes. Cold permafrost is characterized by rapid increased in ground temperature and slow permafrost thawing, whereas warm permafrost undergoes rapid thawing with a slower ground temperature increase (Biskaborn et al., 2019; Smith et al., 2022). This phenomenon can

be attributed to the higher apparent thermal diffusivity in colder permafrost layers, where temperatures near 0 °C led to latent heat consumption by ground ice melts, resulting in lower diffusivity and less energy required to raise ground temperatures (Isaksen et al., 2011; Nicolsky and Romanovsky, 2018).

The cold permafrost tends to be more prevalent in steep mountainous regions in the QTP, as depicted in Figure 6. These areas typically exhibit thin surface sediment layers and shallow bedrock (Shangguan et al., 2017). These characteristics indicate that



the degradation of cold permafrost in the QTP is primarily involves increasing ground temperatures, particularly in regions with limited ground ice. In areas with ice-rich permafrost, the slope effect may contribute to an increased occurrence of thaw slumps (Luo et al., 2022). Notably, approximately half of the warm permafrost area in the QTP is located on flat and gentle slopes (Fig. 6b), where ice-rich permafrost is often found (Zou et al., 2024). Rapid permafrost thawing in these areas may increase surface deformation, resulting in ecological evolution (Jin et al., 2021), hydrology imbalance (Walvoord and Kurylyk,

2016), and engineering stability (Ma et al., 2011). Of particular concern are the Yangtze River and Yellow River basins, where warm permafrost exceeded 80% of the area, are highly susceptible to the impacts of short-term permafrost thawing. The degradation impacts in cool permafrost areas are expected to lies between those in cold and warm permafrost areas and more likely closer to those of the latter due to similar terrain features.

The gridded MAGT$_{15m}$ data generated in this study represents the average value in the period from 2010 to 2019, providing

valuable boundary conditions for future permafrost dynamics investigations in the QTP. For instance, this data can be leveraged in conjunction geothermal datasets to estimate permafrost thickness accurately. Moreover, the depth-fixed MAGT baseline can serve as a robust data for validating numerical model simulations of long-term changes in the permafrost temperature field. These validations are crucial for enhancing our understanding of QTP permafrost responses to environmental drivers and climate change.

**5 Data availability**

The gridded data generated by this study is publicly available and can be downloaded at the National Tibetan Plateau Data Center (TPDC) (https://doi.org/10.11888/Cryos.tpdc.301165, Zou et al., 2024). The data is in GeoTIFF format and can be used with GIS software.

**6 Conclusion**

This study produced gridded data of the mean annual ground temperature at 15 meters depth (MAGT$_{15m}$) during the last decade (2010-2019) at a spatial resolution of nearly 1 km in permafrost regions of the Qinghai-Tibet Plateau (QTP). Regions with negative MAGT$_{15m}$ values covered $1.36\times10^6$ km$^2$ (excluding glacier and lake areas), constituting 44.0% of the QTP area. The average MAGT$_{15m}$ was -1.85 °C (±1.58 °C), with 90% of values in the range of -5.1 °C to -0.1 °C and 51.2% of values were higher than -1.5 °C. The freezing degree days (FDD) was the most significant predictor of MAGT$_{15m}$, followed by the thawing

degree days (TDD), mean annual precipitation (MAP), and soil bulk density (BD). The MAGT$_{15m}$ exhibited a monotonic increase eastward, a slight decrease northward, and an accelerated decrease with increasing elevation across the QTP. Lower MAGT$_{15m}$ values were more prevalent in high mountainous areas with steep slopes. Areas with flat and gentle slopes accounted for approximately half of both the medium (-3.0 to -1.5 °C) and high (-1.5 to 0 °C) MAGT$_{15m}$ intervals. The lowest MAGT$_{15m}$ values were observed in the basins of the Amu Darya, Indus, and Tarim river of the western QTP (-2.7 to -2.9 °C). Conversely,



the Yangtze and Yellow River basins exhibited the highest $MAGT_{15m}$ values (-0.8 to -0.9 °C), and more than 80% of areas were warm permafrost regions. Our gridded $MAGT_{15m}$ dataset can serve as a valuable resource for further studies on deep permafrost characteristics of the QTP, particularly for estimating permafrost thickness.

**Author contributions.** DZ, LZ: conceptualization; DZ, GH, and ED: methodology; GL, WL: data processing and analysis; DZ and CW: original draft writing; LZ and GH: review and editing; LZ: supervision; LZ and GH: funding acquisition.

**Competing interests.** The contact author has declared that none of the authors has any competing interests.

**Acknowledgements.** We would like to express our gratitude to all colleagues and students for their fieldwork contributions in collecting valuable in situ data on the Qinghai-Tibet Plateau. This research was funded by the Second Tibetan Plateau Scientific Expedition and Research (STEP) program (grant no. 2019QZKK0201) and the National Natural Science Foundation of China (grant nos. 41931180 and 42322608).

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
