# Peer review of "Permafrost temperature baseline at the 15 m depth on the Qinghai-Tibet Plateau (2010–2019)"

_Earth System Science Data, 2024_

## Referee Comment (RC1)

[referee-annotated manuscript omitted]

---

## Author Comment (AC1)

**Response to Referee #1**

We appreciate you very much for your comments concerning our manuscript entitled "Permafrost temperature baseline at 15 meters depth in the Qinghai-Tibet Plateau (2010–2019)" (MS No.: essd-2024-114). Those comments are valuable and helpful for improving our manuscript. We followed all comments and made revision and responses carefully. Revised portions are marked in red in the revised manuscript. The line, and figure numbers refer to our revised manuscript. And, a point-by-point reply to the comments are listed below.

**Main comments**

1.  Overall, the paper is well presented based on outstanding field and model works. The data and metadata as well as methodology presented in the paper are very helpful to the geoscientists and engineering in cold regions. We all know that data sharing in permafrost temperature study has been rather difficult. These ground temperature data are thus invaluable in evaluating the thermal state of permafrost and for validating many geocryological, hydrological, ecological and land-surface processes models, and for engineering design and construction in elevational permafrost regions.

    The structures of the paper are well thought out and basically follow the ESSD mandates. However, the authors are encouraged to tell more on the methods of air and ground temperature measurements and their evolutionary paths, since different measurements methods can result in false trends in climate or permafrost changes. For example, your FDD or TDD or your MAGT@DZAA is based on ground or air temperature measurements, and the methods have been advancing rapidly. In the same time, the authors should be more explicit on the criterion selection as why 15 m can be regarded as the DZAA, for which it is evidently illogical. In the meantime, positive MAGT does not necessarily means absence of permafrost because of extensive and increasing presence of supra-permafrost subaerial talik, especially to the east of the QTEC from Golmud-Lhasa and along the engineering lines. Thus, a criterion of subzero MAGT for judging the occurrence of permafrost may underestimate the permafrost extent. That means, you have to be cautious of areas with rapidly or chronically degrading permafrost. Lithology and soil moisture contents are key in defining local or regional DZAA, ALT and MAGT. Thus, using a given depth of either 10 or 15 m as DZAA seems not so reasonable. Thus, if you chose 15 m as the DZAA, you'd better convince readers that your choice is acceptable.

**Response:**

We sincerely appreciate your thorough and insightful review of our manuscript. After carefully considering your comments, we have provided detailed responses to each of the concerns and suggestions you raised.

**1. The measurement methods of FDD/TDD and MAGT and their evolution.**

In early studies, the calculation of Freezing Degree Days (FDD) and Thawing Degree Days (TDD) primarily relied on air temperature (AT). However, the presence of buffering layers, e.g., snow cover and vegetation, introduces significant discrepancies between AT and ground surface temperature (GST) across different permafrost regions. To account for these variations, the relationship between AT and GST is often expressed using n-factors (Riseborough et al., 2008). With the development of remote sensing techniques and ground-based observations, more and more permafrost mapping studies are utilizing GST as a substitute for AT in model inputs, particularly with the widespread application of MODIS land surface temperature (LST) data (Zou et al., 2017; Obu et al., 2019). Additionally, some studies have demonstrated the superiority of GST in simulating the thermal state of deep permafrost (Luo et al., 2018). Therefore, based on the MODIS LST product, this study utilizes ground-measured GST data for calibration to obtain regional GST data, which is then used as input variables for the model. To more clearly express this evolution, we have taken the manuscript context into account and added the following description in the manuscript (Line 121-122):

*"GST, corrected based on MODIS LST, was selected in this study due to its superior performance over air temperature in permafrost modeling (Luo et al., 2018)."*

For the measurement of ground temperature at various depths in the Qinghai-Tibet Plateau (QTP) permafrost regions, are predominantly conducted using thermistors chains assembled by the National Key Laboratory of Frozen Soil Engineering (SKLFSE, CAS). All the $MAGT_{15m}$ data utilized in this study, encompassing both data collected by our own observations and data obtained from the literatures during the 2010-2019, were measured using the same thermistors chains. This methodology is explicitly documented in the cited references. The chains used for these measurements employ thermistor temperature sensors with an accuracy of $\pm 0.05$ °C. These sensors are configured using a cable equipped with a string of thermistors at various depths. The standardized use of the same observational equipment provides a robust foundation for the comparability of $MAGT_{15m}$ data across different regions. We have further elaborated on this in the manuscript as follows (Line 73-74):

*"All ground temperature measurements were obtained using the same equipment, ensuring the comparability of $MAGT_{15m}$ across various permafrost regions."*

**2. Considering 15m as DZAA and the issue of talik.**

We fully agree with your comment that interpreting 15 m as DZAA is evidently illogical. This misunderstanding may have arisen from unclear description in our writing as below:

*"The data of MAGT at 15 m in depth are used for spatialization, considering that DZAAs generally ranges from 10 to 15 m in the QTP (Zhao et al., 2010)."*

However, the intention of our study was not to produce a distribution map of $MAGT_{DZAA}$. To the best of our knowledge, Ran et al. (2021 and 2022) have conducted

several mapping of $MAGT_{DZAA}$ across various regions, including the QTP and the Northern Hemisphere. These studies have made significant contributions to the spatial analysis of $MAGT_{DZAA}$. In contrast, our study aims to provide a fixed-depth deep ground temperature map to facilitate the estimation of permafrost thickness, thereby avoid the spatial variability issues associated with DZAA.

The selection of a depth of 15 m is based on two primary considerations. Firstly, it corresponds with the observed DZAA depth range of 10-15 m in the QTP, with the choice of the lower end of this range aimed at enhancing the stability of ground temperature readings, this particularly beneficial in areas with limited observational depth and data availability. Secondly, this depth corresponds with the extent of existing boreholes, thereby facilitating the integration of a larger dataset into the mapping process.

Although our $MAGT_{15m}$ map exhibits similar spatial patterns to existing $MAGT_{DZAA}$ map, it is fundamentally distinguished by its theoretical framework within permafrost research. The key difference lies in depth: DZAA varies spatially, whereas $MAGT_{15m}$ represents a constant depth.

The adoption of $MAGT_{DZAA}$ as a reference for our $MAGT_{15m}$ mapping is due to the relative scarcity of research on deep permafrost temperatures. $MAGT_{DZAA}$ stands out as one of the few indexes with notable advancements in this area. In contrast, there is a lack of comprehensive studies on deeper ground temperatures based on observed data, largely due to the challenges associated with obtaining such observations. We have undertaken considerable efforts to collect and compile the $MAGT_{15m}$ data during 2010-2019 to support the completion of this study.

To avoid any ambiguity, we have revised the sentence as follows (Line 58-61):

*"This study aims to establish a fixed-depth deep permafrost temperature baseline using data from the QTP for a decade (2010-2019) and a machine learning approach to address the limitations associated with the use of $MAGT_{DZAA}$. Considering the availability of ground temperature records, the data of MAGT at 15 m in depth are used for spatialization."*

*"In the meantime, positive MAGT does not necessarily means absence of permafrost because of extensive and increasing presence of supra-permafrost subaerial talik, especially to the east of the QTEC from Golmud-Lhasa and along the engineering lines. Thus, a criterion of subzero MAGT for judging the occurrence of permafrost may underestimate the permafrost extent. That means, you have to be cautious of areas with rapidly or chronically degrading permafrost."* Your suggestions above indeed highlight one of the key challenges faced in the mapping work of this study. Initially, we considered keeping some positive $MAGT_{15m}$ values to ensure coverage of most permafrost exist regions. However, due to the high variability in geothermal gradients of the permafrost base, determining an appropriate positive MAGT threshold proved challenging. After carefully reviewing your comments, we have followed the conventions of previous studies and retained regions with MAGT < 0.5 °C in the revised

manuscript, to encompass areas where talik is more likely to be widespread. To ensure the reliability of permafrost temperature analysis, we did not reanalyze data with $MAGT_{15m} > 0°C$ in the *Result section* of the revised manuscript. As an alternative, we have included a discussion of regions with $MAGT_{15m} > 0 °C$, as outlined below (Line 255-257):

 *"Additionally, permafrost may still persist in areas where $MAGT_{15m}$ exceeds 0 °C. Statistical analysis reveals that the areas with $MAGT_{15m}$ within the ranges of 0-0.1 °C and 0-0.2 °C cover approximately $0.05×10^6$ $km^2$ and $0.10×10^6$ $km^2$, respectively."*

In addition, ESSD papers should try to avoid over-interpret the patterns or trends of data. It is supposed to tell the integral story of the data structure and functions. Please make the paper concise and on the point, avoiding unnecessary details as possible.

**Response:**

Thank you for your insightful suggestions. To avoid over-interpreting patterns or trends in the data, and after thoroughly reviewing your annotations, we have decided to remove certain sections of the discussion related to permafrost degradation. This adjustment ensures that the manuscript maintains a clear focus on the data structure and functionality. The specific changes are detailed in *Minor Comment 7*.

Other minor issues regarding editing of the MS are provided on the marked MS in the attached document. This is a very quick editing. It is up to authors to ensure the presenting quality to suffice the ESSD standards based on meticulous efforts.

**Response:**

We have thoroughly reviewed and carefully addressed each of your proposed revisions, while also correcting similar issues throughout the manuscript. We greatly appreciate your detailed feedback and constructive insights.

**Minor comments**

1. This sentence is problematic!!!One can hardly imagine that the Tarim Basin and the Amu Darya Basin have lowest MAGTs? Do they belong to the QTP? Are you including the Tianshan Mountains and surrounding basins, as well as the intermontane basins, in the extent of the QTP? Should the title of the paper be changed to that of the Third Pole?

**Response:**

The study area of this research is the Qinghai-Tibet Plateau (QTP), a natural geographical unit that spans both domestic and international regions, excluding the Tianshan Mountains and surrounding basins. The delineation of basins used in this study is derived from research conducted by the Institute of Tibetan Plateau Research,

Chinese Academy of Sciences, and has been categorized into 12 major river basins. Some of these basins encompass only the headwater regions. For improved clarity, we have revised the sentence as follows (Line 20-22):

*"The MAGT$_{15m}$ was the lowest in the headwater areas of the Amu Darya, Indus, and Tarim river basins (-2.9 to -2.7 °C) and the highest in the headwater areas of the Yangtze and Yellow river basins (-0.9 to -0.8 °C)."*

2. Tarim RB is wrongly located. There should be a space between Amu Darya RB

**Response:**

The Tarim river basin is a major basin that includes several smaller sub-basins. In this study, it primarily refers to the headwater areas of the Yarkant River and Hotan river. To improve the figure, we have relocated the label "*Tarim*" to more precisely reflect the Tarim River's headwater region. Additionally, we have updated the label "*Amu Darya*" in *Figure 1*.

[Figure]

***Figure 1:*** *Distribution of boreholes (n=231) for monitoring mean annual ground temperature at 15 m in depth (MAGT15m) on the Qinghai-Tibet Plateau.*

3. Legend items are hard to distinguish because of similar colors and the same shape. Right lower inset, Ground warming rate?

**Response:**

To clearly distinguish the legend items, we have updated *Figure 2* using a darker color palette, ranging from dark blue to dark red, representing the gradient of permafrost temperature from low to high. The right lower inset illustrates the relationship between warming rates and average MAGT$_{15m}$ values.

[Figure]

***Figure 2:*** *Warming rates of MAGT$_{15m}$ during 2010-2019 (a) and the relationship between warming rates and the average MAGT$_{15m}$ (b).*

4. Maybe you should mark out major mountain ranges in the lowest inset.

**Response:**

We greatly appreciate your suggestions. In response, we have marked out the major mountain ranges to facilitate a more precise interpretation of the spatial distribution patterns of MAGT$_{15m}$.

[Figure]

*Figure 5: Variations of mean annual ground temperature at 15m depth (MAGT$_{15m}$) along elevation (a), longitude (b), and latitude (c) transects on the Qinghai-Tibet Plateau (the dashed red line represents the mean MAGT$_{15m}$, and the light-red shaded area indicates its standard deviation; the cyan dashed line shows the areal percentage).*

5. Legend needs unit and name

**Response:**

We have added the legend name and unit in the revised manuscript.

[Figure]

*Figure 7: Distribution (a) and percentage of area in three intervals (b) of MAGT at 15 m depth (MAGT$_{15m}$) in 12 basins of the Qinghai-Tibet Plateau during 2010-2019.*

6. Traditionally and internationally, -1.0C is regarded as the divide. If you use -1.5C, you have to logically persuade readers why would you challenge this criterion. Does Dr. Ran have the clout/right to define warm permafrost, or who else, to deviate from the tradition? Especially, on the QTP or in China, where warm permafrost dominates, why would you define warm permafrost at a lower MAGT? Would that indicate that you do not have enough proportion of cold or warm permafrost? why not -3 or -5C as have done by some Russians and Canadians?

**Response:**

Our initial use of -1.5 °C was aimed at maintaining consistency with the previously defined temperature ranges, facilitating a coherent discussion based on earlier figures and descriptions. After reviewing your feedback in detail, we revisited the literature and found that the -1.5 °C threshold was derived from roadbed deformation assessment studies on the QTP, where Liu et al. (2002) established this threshold based on observed pavement deformation and MAGT. However, for natural permafrost on the QTP, a threshold of -1.0 °C is more commonly used to differentiate between cold and warm permafrost (Wu et al., 2010). In the revised manuscript, we have recalculated the distribution of cold and warm permafrost using -1.0 °C as the threshold and have updated the relevant descriptions accordingly (Line 258-261).

"*Based on the classification criteria established by MAGT$_{DZAA}$, the permafrost can be categorized into cold (≤ -1.0 °C) and warm (> -1.0 °C) permafrost (Wu et al., 2010). Using the predicted MAGT$_{15m}$ data, we analyzed the distribution characteristics of permafrost on the QTP based on this classification. Cold permafrost was the dominant type, covering 63.7% of the permafrost regions, while warm permafrost accounted for 36.3% during the period from 2010 to 2019.*"

7. This is not necessarily true! Warm permafrost may have rich ice content and is slow in thawing, why colder permafrost, generally rocky and found at very high locations. In addition, the thermal inertia would work out, ans apparent head is negligible in comparison with latent heat for ice-rich permafrost regarding the thermal stability.

ibid. This is simply not true. Many locations in Arctic and Boreal zones and on Uplands (of course on the QTP and in Central Asian mountains), permafrost has been persisted at about the freezing temperatures (close to negative zero) for a long, long time, and has not shown the trend of thaw. This is called the zero geothermal degree mode of permafrost degradation or ground temperature curve. Instead of quoting those authors working in regions from other different zones, we may look at more from those working in nearby zones of the QTP.

**Response:**

*"Cold permafrost is characterized by rapid increased in ground temperature and slow permafrost thawing, whereas warm permafrost undergoes rapid thawing with a slower ground temperature increase (Biskaborn et al., 2019; Smith et al., 2022). This phenomenon can be attributed to the higher apparent thermal diffusivity in colder permafrost layers, where temperatures near 0 °C led to latent heat consumption by ground ice melts, resulting in lower diffusivity and less energy required to raise ground temperatures (Isaksen et al., 2011; Nicolsky and Romanovsky, 2018)."*

Considering the incomplete conclusions, regional misalignment, and the style of the ESSD journal (also recommended by the other reviewer), we have decided to remove this section of text. This revision aims to maintain a clear focus on the QTP region and the data structure and functionality presented in the manuscript.

8. Compared to the thickness of the QTP, permafrost, generally a few meters to a kilometer to the best, is very shallow and on the surface of the plateau. In the QTP is generally more often used in geology and geophysics of the QTP studies. For most geographical and geocryological, glacial studies, on the QTP is more proper.

**Response:**

We have expanded on the significance of this study's findings in the context of geology and geophysics research on the QTP as follows (Line 277-278):

*"These validations are crucial for enhancing our understanding of QTP permafrost responses to environmental drivers and climate change. Additionally, the $MAGT_{15m}$ data offers critical insights for understanding geological processes and ecosystem dynamics, thereby supporting related studies in the QTP permafrost regions."*

References:

Riseborough, D., Shiklomanov, N., Etzelmüller, B., Gruber, S., and Marchenko, S.: Recent advances in permafrost modelling, Permafr. Periglac., 19, 137–156, doi:10.1002/ppp.615, 2008.

Zou, D., Zhao, L., Sheng, Y., Chen, J., Hu, G., Wu, T., Wu, J., Xie, C., Wu, X., Pang, Q., Wang, W., Du, E., Li, W., Liu, G., Li, J., Qin, Y., Qiao, Y., Wang, Z., Shi, J., and Cheng, G.: A new map of permafrost distribution on the Tibetan Plateau, Cryosphere, 11, 2527–2542, doi:10.5194/tc-11-2527-2017, 2017.

Obu, J., Westermann, S., Bartsch, A., Berdnikov, N., Christiansen, H. H., Dashtseren, A., Delaloye, R., Elberling, B., Etzelmüller, B., Kholodov, A., Khomutov, A., Kääb, A., Leibman, M. O., Lewkowicz, A. G., Panda, S. K., Romanovsky, V., Way, R. G., Westergaard-Nielsen, A., Wu, T., Yamkhin, J., and Zou, D.: Northern Hemisphere permafrost map based on TTOP modelling for 2000–2016 at 1 km$^2$ scale, Earth-Sci. Rev., 193, 299–316, doi:10.1016/j.earscirev.2019.04.023, 2019.

Luo, D., Jin, H., Marchenko, S. S., and Romanovsky, V. E.: Difference between near-surface air, land surface and ground surface temperatures and their influences on the frozen ground on the Qinghai-Tibet Plateau, Geoderma, 312, 74–85, doi.org/10.1016/j.geoderma.2017.09.037, 2018.

Ran, Y., Li, X., Cheng, G., Nan, Z., Che, J., Sheng, Y., Wu, Q., Jin, H., Luo, D., Tang, Z., and Wu, X.: Mapping the permafrost stability on the Tibetan Plateau for 2005–2015, Sci. China Earth Sci., 64, 62–79, doi:10.1007/s11430-020-9685-3, 2021.

Ran, Y., Li, X., Cheng, G., Che, J., Aalto, J., Karjalainen, O., Hjort, J., Luoto, M., Jin, H., Obu, J., Hori, M., Yu, Q., and Chang, X.: New high-resolution estimates of the permafrost thermal state and hydrothermal conditions over the Northern Hemisphere, Earth Syst. Sci. Data, 14, 865–884, doi:10.5194/essd-14-865-2022, 2022.

Liu, Y., Wu, Q., Zhang, J., and Sheng, Y.: Deformation of highway roadbed in permafrost regions of the Tibetan Plateau, Journal of Glaciology and Geocryology, 01, 10-15, 2002 (in Chinese).

Wu, Q., Zhang, T., and Liu, Y.: Permafrost temperatures and thickness on the Qinghai-Tibet Plateau, Glob. Planet. Change, 72, 32–38, doi:10.1016/j.gloplacha.2010.03.001, 2010.

---

## Author Comment (AC2)

**Response to Referee #2**

We appreciate you for your comments concerning our manuscript entitled "Permafrost temperature baseline at 15 meters depth in the Qinghai-Tibet Plateau (2010–2019)" (MS No.: essd-2024-114). Those comments are valuable in helping us improve the quality of the manuscript. We have carefully addressed all the points raised and revised the manuscript accordingly. Changes made are highlighted in blue in the revised version. Line and figure numbers refer to the updated manuscript, and a detailed point-by-point response to your comments is provided below.

**General comments**

1. Zou et al. present a dataset that extrapolates ground temperatures over the QTP at the depth of zero annual amplitude (here determined to be at 15 m depth). They use a support vector regression to predict ground temperatures based on nine environmental predictors. They justify this approach by claiming that this method has been shown to be superior to other supervised learning algorithms such as random forest in one study (Ran et al., 2021). While the dataset is novel in the sense that no ground temperatures at 15 m depth have been predicted with this method in the QTP, I have a few concerns about the methods used to create the dataset and the fact that a similar dataset exists on a pan-Arctic scale for the entire permafrost region through the permafrost cci ground temperature dataset. Dismissing this dataset solely on the grounds of it not reaching as deep as the dataset presented in this study is not sufficient in my opinion. Especially considering the fact that the authors claim that the DZAA ranges from 10 to 15 m in central Asia and therefore would partially be covered by the permafrost cci product. Furthermore, the R2 value of the prediction is below 0.5, meaning that less than half of the variance in ground temperature can be explained by the model. This suggests that the model could potentially be improved or a different model should be tested to see if the predictions accuracy can be increased.

**Response:**

We fully agree with your comment that interpreting 15 m as DZAA is evidently illogical, which was also pointed out by the other reviewer. This misunderstanding may have resulted from unclear description in our writing, as outlined below:

"*The data of MAGT at 15 m in depth are used for spatialization, considering that DZAAs generally ranges from 10 to 15 m in the QTP (Zhao et al., 2010).*"

However, the objective of our study was not to generate a map of MAGT$_{DZAA}$. To our knowledge, Ran et al. (2021, 2022) have already conducted comprehensive mapping of MAGT$_{DZAA}$ across various regions, including the QTP and the Northern Hemisphere, significant contributing to the spatial analysis of MAGT$_{DZAA}$. In contrast, our study aims to provide a fixed-depth deeper ground temperature map to support the permafrost thickness estimation, thereby avoiding the spatial variability challenges

inherent to DZAA.

The selection of a depth of 15 m is based on two primary considerations. Firstly, it corresponds with the observed DZAA depth range of 10-15 m in the QTP, with the choice of the lower end of this range aimed at enhancing the stability of ground temperature readings, this particularly beneficial in areas with limited observational depth and data availability. Secondly, this depth corresponds with the extent of existing boreholes, thereby facilitating the integration of a larger dataset into the mapping process.

Therefore, the $MAGT_{15m}$ presented in this study does not overlap with the CCI products. In terms of depth, the CCI data provides permafrost temperatures at a maximum depth of 10 m, while our study presents data at a depth of 15 m. Additionally, the CCI products cover the permafrost regions of the Northern Hemisphere north of 30 °N, excluding the southernmost permafrost areas of the QTP.

Although our $MAGT_{15m}$ map exhibits similar spatial pattern to existing $MAGT_{DZAA}$ maps, it is fundamentally distinguished by its theoretical framework within permafrost research. The primary difference lies in depth: DZAA varies spatially, whereas $MAGT_{15m}$ map represents a fixed depth.

The adoption of $MAGT_{DZAA}$ as a reference and introduction for our $MAGT_{15m}$ mapping stems from the relative scarcity of research on deep permafrost temperatures. $MAGT_{DZAA}$ stands out as one of the few indexes with significant advancements in this area. In contrast, studies on deeper ground temperatures based on observed data are lacking, primarily due to the challenges of acquiring such observations. We have made substantial efforts to collect and compile the $MAGT_{15m}$ data for the period 2010-2019 to support the completion of this study.

To avoid any ambiguity, we have revised the sentence as follows (Line 58-61):

*"This study aims to establish a fixed-depth deep permafrost temperature baseline using data from the QTP for a decade (2010-2019) and a machine learning approach to address the limitations associated with the use of $MAGT_{DZAA}$. Considering the availability of ground temperature records, the data of MAGT at 15 m in depth are used for spatialization."*

The lower $R^2$ of the $MAGT_{15m}$ predictions in comparison to $MAGT_{DZAA}$ may be attributed to the greater depth of temperature prediction, in the context of same method and similar environmental variables. Observations reveal that DZAA in the QTP predominantly occurs at depths shallower than 15 m, especially in areas close to the permafrost boundary, where DZAA are often even shallower. For instance, in the Xidatan area, located at the northern boundary of the QTP, the DZAA is recorded to be approximately 5 to 7 meters (Liu et al., 2021). DZAA represents the maximum depth that seasonal surface temperature fluctuations can reach, and the $MAGT_{DZAA}$ values are closely related to the climatic conditions of nearby years. Utilizing contemporary or recent air temperature or ground surface temperature data (such as FDD and TDD) in predicting spatial distributions generally yields higher $R^2$ values.

In comparison, the MAGT$_{15m}$, due to its greater depth, is more closely linked to long-term climatic conditions, as the propagation of temperature exhibits a lag effect. In other words, the increased depth of the strata is likely the primary factor contributing to the lower $R^2$ of the MAGT$_{15m}$ predictions. Nevertheless, a significant relationship exists between the predicted and observed MAGT$_{15m}$ values (p<0.001) in this study, and both bias and RMSE, along with their standard deviations, are slightly lower than those reported in previous studies. Considering these thermal propagation characteristics, we have extended the periods for FDD and TDD to 2003-2019 and have calibrated these metrics based on observed GST data to enhance the representativeness of surface temperature variables.

I do not want to dismiss the work that the authors have put into this dataset, however I am unsure if it offers a significant contribution to the scientific community in its current state. I have a few suggestions on how to enhance the impactfulness of the paper, but I am unsure if it then still fits the scope of ESSD.

1.The SVR method has been tested by Ran et al., 2021 and found to be sufficient for their purposes. However, their R2 was 0.71 as compared to 0.48 in this study. Further, they have tested various different supervised learning algorithms to conclude that SVR is the best model to use, which is lacking in the present manuscript. Hence, I would suggest the authors also perform a test for the other models in question that can be used for this task to get a better idea of their individual performance.

**Response:**

Before the initial submission of this manuscript, we had already tested the methods proposed by Ran et al. (2021). The results of these methods exhibited significant differences, both in terms of statistical metrics and spatial patterns. *Table R1* presents the performance of four statistical models.

In terms of $R^2$, the random forest (RF) model performs the best with a 0.92 value, and the $R^2$ of the generalized linear model (GLM) and generalized additive model (GAM) being comparable to that of the support vector regression (SVR) model (0.47-0.48). For bias and RMSE, the RF model shows the lowest values; the RMSE of the SVR method is slightly lower than those of the GLM and GAM. From the performance (*Table R1*), the RF is undoubtedly the best model. However, examining the spatial pattern of the RF-predicted MAGT$_{15m}$ (*Fig. R1.b*), most values in the permafrost regions are concentrated between -3 °C and -0.5 °C, with a minimum of only -3.2 °C, which is not consistent with observational facts. In addition, in several seasonally frozen ground regions, such as the Qaidam Basin and southern endorheic zones, the predicted MAGT$_{15m}$ falls below 0°C, suggesting the presence of permafrost, which contradicts existing permafrost distribution maps. The results produced by the RF model may be attributed to overfitting to the observational dataset. Although parameter adjustments can enhance certain aspects of the spatial pattern of MAGT$_{15m}$, the overall outcomes still fall short of expectations.

For linear-type models such as GLM and GAM, their performance is comparable to that of SVR. However, the predicted $MAGT_{15m}$ values often exhibit a seesaw effect, where lower values are predicted in high mountain areas and higher values at the permafrost margins. This seesaw effect becomes more pronounced when fewer variables are selected following collinearity analysis. After comparing the model performances and spatial patterns of different methods, we ultimately selected the SVR model for predicting $MAGT_{15m}$ in this study. Moreover, SVR is a deterministic prediction method, ensuring consistent and reproducible results with a fixed set of sample points. This choice aims to establish a methodological foundation for future analyses involving the addition of more sample points and comparisons across different input datasets.

**Table R1:** *Predictive performance of mean annual ground temperature at 15 m in depth ($MAGT_{15m}$) for four statistical models[*].*

| Performance | SVR | RF | GLM | GAM |
| --- | --- | --- | --- | --- |
| $R^2$ | 0.48 (±0.14) | 0.92 (±0.03) | 0.47 (±0.13) | 0.48 (±0.14) |
| Bias (°C) | -0.01 (±0.11) | -0.00 (±0.05) | 0.01 (±0.12) | 0.01 (±0.13) |
| RMSE (°C) | 0.71 (±0.13) | 0.32 (±0.05) | 0.72 (±0.12) | 0.72 (±0.12) |

[*]*SVR, support vector regression; RF, random forest; GLM, generalized linear model; GAM, generalized additive model. $R^2$, bias, and RMSE with 1 standard deviation.*

[Figure]

**Figure R1:** *Spatial distribution of predicted mean annual ground temperatures at the 15m depth ($MAGT_{15m}$) across the Qinghai-Tibet Plateau during 2010-2019, based on support vector regression (a) and random forest (b) models.*

2. Currently, the dataset is presented as a stand-alone dataset to be published in ESSD. However, the overlap with the existing permafrost cci ground temperature dataset can not be denied. My suggestion may significantly change the scope of the paper, but I wonder if it would make more sense to use the borehole data used in this study to assess how useful the ground temperatures could be to inform e.g., boundary conditions of permafrost models in the QTP. As the authors describe, the boreholes are equipped with thermistor strings, which probably means that measurements are available at several depths. This would serve as a basis to compare the borehole data directly to the ground temperature dataset at 10 m depth. A comparison to the existing dataset could then be a better motivation to conduct your own supervised learning method to improve the accuracy. However, if the $R^2$ is similar or higher when directly compared to the existing data (permafrost cci), there may not be a need for this since depth extrapolations of temperatures below the DZAA could be achieved with geothermal heat flux and simpler heat conduction models.

Regardless of the scope of the final manuscript, I think a comparison to the existing datasets is crucial, considering the model in this study explains a relatively low amount of variance in the data.

**Response:**

As previously mentioned, the goal of this study is to map the fixed-depth deep ground temperature of permafrost on the QTP. Based on advancements in deep ground temperature research ($MAGT_{DZAA}$) and the availability of existing dataset, we selected 15 m as the mapping depth. This choice differs significantly from the CCI data, both in terms of depth and geographic coverage of the QTP. As an independent dataset, our results focus on the permafrost temperature at a depth of 15 m, which can serve as an upper boundary condition for future studies on deeper permafrost characteristics.

Due to the inability to establish a strict correspondence in depth, it is not appropriate to directly compare the results of this study with the CCI ground temperature data. While comparing borehole data with the existing 10 m depth CCI dataset is a valuable suggestion, it is somewhat outside the scope of this manuscript's objectives. However, we will consider conducting a separate evaluation in future research. Thank you very much for your insightful comments.

Modeling the regional thermal dynamics of permafrost beneath the DZAA remains to pose significant challenges for thermal conduction models. A major difficulty lies in assessing simulation uncertainty, which is one of the key motivations for adopting a fixed depth of 15 m for spatialization of ground temperature in this study. Our objective is to establish a baseline using observational data that can facilitate the comparison and evaluation of results produced by thermal conduction models.

Specific comments:

1. L56: What kind of datasets are you talking about here? Either delete the last part of

the sentence or give an overview (for example in a table) about the datasets you are talking about here.

**Response:**

We have added specific dataset name "ground temperature", and the revision is as follows (Line 56-57):

"*Over the past two decades, permafrost monitoring efforts on the QTP have established a substantial monitoring network and ground temperature datasets have been published (Zhao et al., 2021).*"

2. L75: Do I understand correctly that you implemented a procedure to fill temporal gaps in 78% of the data based on 22% of the observations? Please clarify.

**Response:**

Of the monitoring sites, 22% have maintained continuous observations over multiple years. Before establishing the relationships, we assessed an evaluation of these sites, which revealed that the $MAGT_{15m}$ range for this dataset was from -3.95 °C to 0.03 °C. This range effectively captures the essential spectrum of permafrost ground temperatures across the QTP and closely aligns with the observed thermal characteristics of permafrost in the region.

3. L77-82: From what I understand, you used 51 sites to calculate a linear trend to fill the gaps in the remaining 180 sites by assuming they all experience the same warming trend. However, your Fig. 2a clearly shows that warming trends are very different for cold vs. warm permafrost. I think applying a single warming trend that is based on 22% of the data is very problematic here. If I misunderstood this part, please clarify. Otherwise I am doubtful of the reliability of this preprocessing step.

**Response:**

We sincerely appreciate your detailed review; it is crucial to clarify that our methodology does not rely on a single warming trend for filling missing values. As demonstrated in *Fig. 2a*, there are significant differences in the warming rates between cold and warm permafrost; in general, cold permafrost tends to exhibit a more rapid warming rate, whereas warm permafrost warms at a comparatively slower rate. *Fig. 2b* illustrates the established relationship between $MAGT_{15m}$ values, organized from low to high temperature, and their corresponding warming rates. Based on this relationship, we subsequently calculated the warming rates for various monitoring sites using the observed $MAGT_{15m}$ values.

[Figure]

*Figure 2: Warming rates of $MAGT_{15m}$ during 2010-2019 (a) and the relationship between warming rates and the average $MAGT_{15m}$ (b).*

4. Eq 1. An $R^2$ of 0.45 does not create a lot of trust into your interpolation method (see comment above).

**Response:**

Although the $R^2$ value of 0.45 is relatively modest, statistical analysis reveals that the relationship between predicted and observed $MAGT_{15m}$ values is highly significant ($p<0.001$). At present understanding, the magnitude of $MAGT_{15m}$ is the dominant factor controlling the warming rate of $MAGT_{15m}$. However, in addition to $MAGT_{15m}$, the warming rate may also be closely related to permafrost characteristics (e.g., soil texture and ground ice content) and active layer properties (e.g., soil moisture and active layer thickness), as well as the magnitude of climate change. At this stage of the research, given the lack of more detailed or accurate site-specific observations of permafrost and its environmental characteristics, we primarily attribute the variations in the warming rate to differences in $MAGT_{15m}$.

5. L107: I am not very familiar with SVR, but is a 90/10 a typical split for this method? I was expecting a 80/20 or even a 70/30 split since you do not have a very large dataset. Can you provide the model performances with different splits? And how high is the risk for overfitting with the 90/10 split?

**Response:**

In the SVR method, a 90/10 split ratio is commonly used, as referenced in Ran et al.

(2021), and determined based on the sample size of this study. Considering your suggestion, we further evaluated model performance using 80/20 and 70/30 split ratios, as shown in *Table R2*. The $R^2$ and RMSE across the three split ratios (90/10, 80/20, and 70/30) ranged from 0.45 to 0.48 and 0.71 to 0.73, respectively, with a bias of -0.01 in all cases. These results indicate that there are no significant variances between the three split ratios when using the SVR method, thereby supporting the validity of the 90/10 split.

***Table R2:*** *Predictive performance of the support vector regression (SVR) model across various split ratios.*

| Split ratio (%) | $R^2$ | Bias (°C) | RMSE (°C) |
|---|---|---|---|
| 90/10 | 0.48 (±0.14) | -0.01 (±0.11) | 0.71 (±0.13) |
| 80/20 | 0.46 (±0.09) | -0.01 (±0.07) | 0.72 (±0.08) |
| 70/30 | 0.45 (±0.07) | -0.01 (±0.06) | 0.73 (±0.07) |

6. L132: "high accuracy" is inappropriate here. How do you determine it is "high"? The indicators you are describing are not creating a lot of confidence.

**Response:**

The term "high accuracy" was inappropriate, as you suggested, we have removed the relevant description, as follows (Line 137-139):

*"The cross-validation of 1000 runs demonstrated that the mean values of the three statistical indicators, i.e., bias, root-mean-square error (RMSE), and coefficient of determination ($R^2$) were -0.01 °C (±0.11 °C), 0.71 °C (±0.13 °C), and 0.48 (±0.14), respectively."*

7. Fig. 3: Please add a label for the red line either in the figure or in the caption.

**Response:**

We have added the label for the red line in *Fig.3* as per your suggestion.

[Figure]

*Figure 3:* *Relationship between predicted and observed mean annual ground temperatures at 15 m depth (MAGT$_{15m}$) in permafrost regions on the Qinghai-Tibet Plateau during 2010-2019.*

8. Fig. 4: Maybe I have missed it in the text with all the numbers, but did you say that you are masking all values > 0°C? It looks like the final dataset only shows values < 0°C. Is that because you do not have confidence in non-frozen conditions? Are you assuming that there is no permafrost in regions with T > 0°C? Please clarify this throughout your results section.

**Response:**

In *Fig. 4* of the original manuscript, we have presented only values > 0°C to highlight the MAGT$_{15m}$ in the permafrost region of the QTP, which is the most significant result of this study. To show the prediction results for areas with positive temperatures, we included the complete set of predictions for the entire QTP in *Fig. R2*.

[Figure]

***Figure R2:*** *Spatial distribution of predicted mean annual ground temperatures at the 15m depth across the Qinghai-Tibet Plateau.*

Positive MAGT$_{15m}$ does not necessarily means absence of permafrost because of extensive and increasing presence of supra-permafrost subaerial talik, especially to the east of the QTEC from Golmud-Lhasa and along the engineering lines. Thus, a criterion of subzero MAGT$_{15m}$ for judging the occurrence of permafrost may underestimate the permafrost extent. We have considered keeping some positive MAGT$_{15m}$ values to ensure coverage of most permafrost exist regions. However, due to the high variability in geothermal gradients of the permafrost base, determining an appropriate positive MAGT threshold proved challenging. After carefully reviewing both your comments and those of the other reviewer, we have followed the conventions of previous studies and retained regions with MAGT$_{15m}$ < 0.5 °C in the revised manuscript, to encompass areas where talik is more likely to be widespread. To ensure the reliability of permafrost temperature analysis, we did not reanalyze data with MAGT$_{15m}$ > 0 °C in the *Result section* of the revised manuscript. As an alternative, we have included a discussion of regions with MAGT$_{15m}$ > 0 °C, as outlined below (Line 255-257):

*"Additionally, permafrost may still persist in areas where MAGT$_{15m}$ exceeds 0 °C. Statistical analysis reveals that the areas with MAGT$_{15m}$ within the ranges of 0-0.1 °C and 0-0.2 °C cover approximately 0.05×10⁶ km² and 0.10×10⁶ km², respectively."*

[Figure]

***Figure 4:*** *Spatial distribution of predicted mean annual ground temperatures at the 15m depth (MAGT$_{15m}$) across the Qinghai-Tibet Plateau during 2010-2019.*

9. Section 3.2.2: This section is very difficult to read. Would it be possible to put all those numbers into a table, refer to it in the text and focus on the conceptual characteristics only?

**Response:**

In the classification system based on MAGT$_{DZAA}$, permafrost can be divided into

three types: cold (≤ -3.0 °C), cool (-3 to -1.5 °C), and warm (> -1.5 °C) permafrost (Ran et al., 2022). However, there are significant differences in both depths and values between $MAGT_{15m}$ and $MAGT_{DZAA}$, and using this classification system in the *Results* section may lead to confusion. Although we have not placed all the relevant numbers into a single table then directly referenced their conceptual characteristics, we have made efforts to simplify the numerical descriptions to enhance the readability of the text of the *Section 3.2.2*.

10. Fig 7.: What are the units in the figure legend? I assume °C?

**Response:**

We have added the legend name and unit (°C) in the revised manuscript.

[Figure]

*Figure 7: Distribution (a) and percentage of area in three intervals (b) of MAGT at 15 m depth ($MAGT_{15m}$) in 12 basins of the Qinghai-Tibet Plateau during 2010-2019.*

11. L259-261: This sentence is very confusing and I am not able to follow it. Please see Biskaborn et al., 2019, which you are already citing, for an example on how to describe the difference between warming of "cold" and "warm" permafrost and how it relates to latent heat consumption.Might be worth citing Gruber 2012 (cited later in paper) in your introduction where you discuss TP permafrost maps.

**Response:**

Considering the incomplete conclusions, regional misalignment, and the style of the ESSD journal (also recommended by the other reviewer), we have decided to remove this section of text. This revision aims to maintain a clear focus on the QTP region and the data structure and functionality presented in the manuscript.

References:

Liu, G., Xie, C., Zhao, L., Xiao, Y., Wu, T., Wang, W., and Liu, W.: Permafrost warming near the northern limit of permafrost on the Qinghai-Tibetan Plateau during the period from 2005 to 2017: A case study in the Xidatan area, Permafr. Periglac., 32, 323–334, doi:10.1002/ppp.2089, 2021.

Ran, Y., Li, X., Cheng, G., Nan, Z., Che, J., Sheng, Y., Wu, Q., Jin, H., Luo, D., Tang, Z., and Wu, X.: Mapping the permafrost stability on the Tibetan Plateau for 2005–2015, Sci. China Earth Sci., 64, 62–79, doi:10.1007/s11430-020-9685-3, 2021.

Ran, Y., Li, X., Cheng, G., Che, J., Aalto, J., Karjalainen, O., Hjort, J., Luoto, M., Jin, H., Obu, J., Hori, M., Yu, Q., and Chang, X.: New high-resolution estimates of the permafrost thermal state and hydrothermal conditions over the Northern Hemisphere, Earth Syst. Sci. Data, 14, 865–884, doi:10.5194/essd-14-865-2022, 2022.

---

## Author Response (AR2)

**Response to Referee #2 for the Second review**

We sincerely appreciate your second review and the insightful comments regarding our manuscript entitled "Permafrost temperature baseline at 15 meters depth in the Qinghai-Tibet Plateau (2010–2019)" (MS No.: essd-2024-114). Your feedback has been invaluable in improving the quality of the manuscript. We have thoroughly addressed all the points raised and have made the necessary revisions accordingly.

**General comments**

I would like to acknowledge the effort the authors have put into improving the manuscript according to both reviewers' comments. They have clearly taken the time to consider concerns and redo analysis to strengthen their study.

Unfortunately, I am concerned about the decisions made in terms of the appropriate machine learning algorithm. I appreciate that the authors have added the results from the other viable machine learning models they mention in the original manuscript, but I struggle with the decision and the arguments as to why the much better performing random forest model was dismissed. As I am not an expert in machine learning myself, I do not feel qualified to provide in-depth feedback about machine learning algorithms and their performance metrics, but if the statistics used to evaluate the model are R2, bias, and RMSE, I cannot understand how the SVR can be chosen as the best model when it's clearly performing poorly compared to the random forest algorithm.

While I understand the qualitative reasoning and I believe that the authors know the area better than me, I do not think that this alone justifies the choice of model. Are there any other statistics or performance metrics that you could use to evaluate the model and back up your qualitative results numerically? If you say you do not trust the RF model because of the reasons given in your response letter, how can you be sure that a much worse performing algorithm gives you better results?

I further wonder if the results of testing the other machine learning algorithms should be included in the paper as it is crucial for full transparency and reproducibility. Again,

not being an expert in machine learning, I am unsure about the best way to do so, but in its current state, the decisions made by the authors are not fully transparent and comprehensible.

I am happy to take another look at the manuscript after this pressing issue has been addressed. To do so, it would also be very helpful to have a full track changes version of the manuscript, including deleted/changed text between the two manuscripts. Right now I can only see additions and they come in two different colors, which is confusing.

I am aware that my comments may mean a lot of extra work for the authors, but I think it is important for the transparency and reproducibility of your results and its translation into future work and different regions in the permafrost area.

**Response:**

In predicting $MAGT_{15m}$ in this study, the Random Forest (RF) method highlights a common issue where the model's performance may not align with the geographic significance of the results. Despite RF shows a high $R^2$, its predicted values deviate from the actual physical significance due to the narrow distribution range of $MAGT_{15m}$ values, with a minimum of only -3.2 °C and a maximum of 0.9 °C. This limitation arises from RF's over-reliance on the range of the training data and its lack of extrapolation capability. As an ensemble algorithm based on decision trees, RF is adept at capturing complex nonlinear relationships within the data. However, its reliance on splitting rules may result in insufficient predictive power when faced with extreme values or boundary conditions.

RF tends to be highly dependent on the specific distribution of the training data, which makes it susceptible to being influenced by local data structures while neglecting broader geographic trends in ground temperature. This may explain the narrow range in the $MAGT_{15m}$ predictions. In other words, the performance of RF is largely depended on the representativeness of the training samples.

Figure R1 compares the elevation distribution of borehole locations used in the

manuscript with that of the Qinghai-Tibet Plateau (QTP) permafrost regions. Specifically, the comparison reveals that $MAGT_{15m}$ samples are primarily concentrated in areas below 5200 m. Due to challenging environmental conditions and limited access to high-altitude regions, sample points are sparse at elevations above this threshold. This makes it crucial to rely on models that can extrapolate $MAGT_{15m}$ values to these higher regions. However, the RF method demonstrates limited capability in this regard.

A comparison shows that the RF-predicted $MAGT_{15m}$ range (-3.2 to 0.9 °C) closely aligns with the observed range (-4.0 to 1.5 °C). This reflects a characteristic of the RF method: while it excels at modeling local patterns, it is strongly influenced by the available observations and fails to capture variations in extremely low ground temperatures. Based on the observed lapse rate of $MAGT_{15m}$ (0.4-0.9°C/100m) in the QTP permafrost regions, the $MAGT_{15m}$ at 6000 m is conservatively estimated to range from -6 °C to -10 °C. Since regions above 5200 m account for 22.0 % of the QTP permafrost areas, accurate predictions for these higher-altitude regions are crucial for a comprehensive analysis.

The support vector regression (SVR) method, on the other hand, predicts an extremely low value of -12.2 °C, which aligns well with the lapse rate-based estimate. The strength of SVR lies in its ability to capture global trends by identifying an optimal hyperplane in a high-dimensional space. SVR tends to model overarching trends in the data rather than focusing on local variations. While its $R^2$ value is somewhat lower, SVR effectively captures the spatial trends, which is a key advantage for ground temperature predictions in the QTP, especially when accounting for temperature gradients and spatial variations at high altitudes.

[Figure]

*Figure R1:* *Percentage frequency distribution of elevations for $MAGT_{15m}$ borehole locations compared to the permafrost regions of the Qinghai-Tibet Plateau.*

To further explore how different parameters in the RF method influence prediction results, we designed a series of sensitivity experiments. Three key parameters were identified in the RF method: ***mtry***, ***ntree***, and ***maxnodes***.

The ***mtry*** parameter defines the number of predictor variables randomly sampled to be considered for each split in a decision tree. It influences the diversity of the individual trees in the forest. A smaller value increases the model's variance, as it allows trees to explore different splits based on a smaller set of features. Conversely, a larger value leads to more similar trees, reducing variance but potentially increasing overfitting. For regression problems, it is typically set to one-third of the total features. In this study, *mtry* was set to 3, given that there are 10 predictor variables in total.

The ***ntree*** parameter specifies the number of decision trees to grow in the RF, essentially controlling the size of the forest. A larger value generally improves model stability and prediction accuracy by averaging over a greater number of trees. However, this also increases computation time and memory usage. The trade-off lies between

model accuracy and computational efficiency. Typical values often range from 500 to 1000, as seen in previous studies (Ran et al., 2021, 2022). To assess the impact of *ntree* on model performance, we varied it across a range of values: 20, 50, 200, 500, 1000, 2000, and 5000, evaluating its effects on accuracies and the spatial patterns of the predicted results.

The ***maxnodes*** parameter limits the maximum number of terminal nodes (leaves) that each individual tree in the forest can have, controlling the depth and complexity of the trees. A smaller value results in simpler trees with fewer splits, which can reduce overfitting and improve the model's generalization ability, but may introduce bias. Larger values allow trees to grow deeper and capture more complex patterns, but can lead to overfitting, especially with smaller datasets. The optimal value depends on the data and should be fine-tuned through cross-validation. Typically, values between 20 and 50 are chosen. To evaluate the RF model's performance under extreme conditions, we manually set *maxnodes* to 10, 20, 30, 40, 50, 100, and 200.

Table R1 summarizes the $R^2$ values of the RF model for 49 different combinations of *ntree* and *maxnodes* parameters. Each $R^2$ was evaluated using a 9:1 data split, with results averaged over 200 runs to ensure robustness. The analysis shows that parameter selection significantly impacts model performance, with $R^2$ values ranging from 0.66 to 0.92. With *ntree* held constant, increasing *maxnodes* (from 10 to 200) notably improved $R^2$ values. In contrast, when *maxnodes* was held constant, increasing *ntree* (from 20 to 5000) resulted in only a slight improvement in $R^2$. For RMSE, the values across the 49 parameter combinations ranged from 0.31 to 0.60 (Table R2). Increasing *maxnodes* notably reduced RMSE values, while increasing *ntree* led to only a slight reduction in RMSE.

***Table R1:*** *$R^2$ statistics of the Random Forest (RF) model for various combinations of ntree and maxnodes parameters.*

| ntree | maxnodes | | | | | | |
|---|---|---|---|---|---|---|---|
| | 10 | 20 | 30 | 40 | 50 | 100 | 200 |

| | 10 | 20 | 30 | 40 | 50 | 100 | 200 |
|---|---|---|---|---|---|---|---|
| 20 | 0.66 | 0.76 | 0.82 | 0.86 | 0.89 | 0.90 | 0.90 |
| 50 | 0.69 | 0.78 | 0.84 | 0.87 | 0.89 | 0.92 | 0.91 |
| 200 | 0.70 | 0.79 | 0.84 | 0.88 | 0.90 | 0.92 | 0.92 |
| 500 | 0.69 | 0.78 | 0.84 | 0.87 | 0.90 | 0.92 | 0.92 |
| 1000 | 0.69 | 0.78 | 0.84 | 0.88 | 0.90 | 0.92 | 0.92 |
| 2000 | 0.68 | 0.80 | 0.84 | 0.88 | 0.90 | 0.92 | 0.92 |
| 5000 | 0.69 | 0.78 | 0.84 | 0.88 | 0.90 | 0.92 | 0.92 |

*Table R2:* *RMSE statistics of the Random Forest (RF) model for various combinations of ntree and maxnodes parameters.*

| ntree | maxnodes | | | | | | |
|---|---|---|---|---|---|---|---|
| | 10 | 20 | 30 | 40 | 50 | 100 | 200 |
| 20 | 0.60 | 0.50 | 0.44 | 0.41 | 0.36 | 0.35 | 0.34 |
| 50 | 0.60 | 0.50 | 0.43 | 0.39 | 0.36 | 0.31 | 0.33 |
| 200 | 0.58 | 0.49 | 0.43 | 0.37 | 0.35 | 0.31 | 0.32 |
| 500 | 0.59 | 0.50 | 0.43 | 0.39 | 0.35 | 0.32 | 0.31 |
| 1000 | 0.58 | 0.50 | 0.43 | 0.38 | 0.36 | 0.32 | 0.32 |
| 2000 | 0.59 | 0.49 | 0.43 | 0.39 | 0.35 | 0.31 | 0.31 |
| 5000 | 0.59 | 0.49 | 0.42 | 0.38 | 0.36 | 0.32 | 0.32 |

Variations in parameter combinations lead to differences in evaluation metrics. Nonetheless, even the lowest $R^2$ (0.66) and highest RMSE (0.60 °C) achieved by the RF method outperform those of SVR method ($R^2$=0.48, RMSE=0.71). From an accuracy evaluation perspective, the RF method demonstrates robust reliability and offers further potential for improvement through parameter optimization. However, since the RF method is constrained by the distribution of sample points and fails to accurately predict lower $MAGT_{15m}$ values, we are also interested in exploring whether different parameter choices could help address this limitation.

To this end, we analyzed the impact of 49 parameter combinations on the maximum (Table R3) and minimum (Table R4) MAGT$_{15m}$ predictions for the QTP permafrost regions. The results reveal that parameter variations have minimal impact on the extreme values, with maximum predictions consistently ranging from 0.1 °C and 1.1 °C and minimum predictions spanning -2.5 °C to -3.4 °C. These findings suggest that even with optimized parameter settings, the RF method fails to generate reasonable spatial predictions based on the existing sample data.

*Table R3: Maximum value statistics of the Random Forest (RF) model for various combinations of ntree and maxnodes parameters.*

| ntree | maxnodes | | | | | | |
|---|---|---|---|---|---|---|---|
| | 10 | 20 | 30 | 40 | 50 | 100 | 200 |
| 20 | 0.32 | 0.33 | 0.75 | 1.11 | 0.95 | 0.96 | 0.87 |
| 50 | 0.30 | 0.63 | 0.87 | 0.84 | 0.82 | 0.84 | 0.98 |
| 200 | 0.10 | 0.56 | 0.75 | 0.82 | 0.91 | 0.82 | 0.86 |
| 500 | 0.17 | 0.55 | 0.71 | 0.75 | 0.86 | 0.85 | 0.87 |
| 1000 | 0.17 | 0.54 | 0.73 | 0.81 | 0.85 | 0.84 | 0.84 |
| 2000 | 0.16 | 0.55 | 0.71 | 0.80 | 0.85 | 0.87 | 0.86 |
| 5000 | 0.15 | 0.53 | 0.71 | 0.79 | 0.82 | 0.85 | 0.85 |

*Table R4: Minimum value statistics of the Random Forest (RF) model for various combinations of ntree and maxnodes parameters.*

| ntree | maxnodes | | | | | | |
|---|---|---|---|---|---|---|---|
| | 10 | 20 | 30 | 40 | 50 | 100 | 200 |
| 20 | -2.75 | -3.10 | -3.32 | -3.16 | -3.42 | -3.26 | -3.33 |
| 50 | -2.75 | -2.91 | -2.98 | -3.04 | -3.22 | -3.01 | -3.08 |
| 200 | -2.51 | -2.91 | -2.99 | -3.04 | -3.07 | -3.12 | -3.01 |
| 500 | -2.60 | -2.86 | -3.03 | -3.14 | -3.06 | -3.12 | -3.12 |

| 1000 | -2.59 | -2.85 | -3.09 | -3.09 | -3.06 | -3.09 | -3.16 |
| 2000 | -2.59 | -2.86 | -2.98 | -3.08 | -3.10 | -3.11 | -3.14 |
| 5000 | -2.60 | -2.88 | -3.03 | -3.05 | -3.10 | -3.13 | -3.12 |

The RF method is not unique in its ability to capture local patterns. Further testing revealed that other commonly used machine learning algorithms, such as eXtreme Gradient Boosting (XGBoost) and K-Nearest Neighbors (KNN), exhibit similar behavior to the RF method (Figure R2). Under the same data samples, the minimum and maximum values predicted by XGBoost and KNN methods range from -3.3 to -4.0 °C and from 0.9 to 1.5 °C, respectively.

Like RF, both XGBoost and KNN rely on the distribution of the training data for modeling, and their predictions are primarily shaped by the patterns present within the existing data. Whether through the ensemble of decision trees in RF, the gradient boosting mechanism in XGBoost, or the neighbor-based approach in KNN, these models struggle to predict values beyond the sample range. The prediction results of these algorithms generally do not extend outside the distribution range of the training data, particularly when predicting extreme values (either minimum or maximum). These models tend to replicate the boundaries of the training data, making it difficult to accurately forecast values outside of this range. The primary objective of these models is to fit the training data, rather than extrapolate. While they optimize performance by minimizing training error, they lack inherent mechanisms for extrapolation beyond the data distribution.

[Figure]

***Figure R2:*** *Spatial distribution of predicted mean annual ground temperatures at the 15m depth (MAGT₁₅ₘ) across the Qinghai-Tibet Plateau during 2010-2019, based on support vector regression (a), extreme gradient boosting (b), random forest (c) and K-nearest neighbors (d) models.*

In summary, the limited number of sample points in higher-altitude regions, where $MAGT_{15m}$ values are generally lower, prevents the RF method from accurately capturing spatial trends in these areas based solely on the existing dataset. Due to the harsh climatic conditions at high altitudes, obtaining observational data in these regions remains challenging with current observational capabilities. As a result, there is still a need to leverage models capable of extrapolation to estimate ground temperature trends in these high-altitude areas.

In response to the question, "***Are there any other statistics or performance metrics that you could use to evaluate the model and back up your qualitative results numerically?***" — addressing this issue poses a considerable challenge. Due to the limited availability of additional observational data and reliable spatial information, identifying alternative statistical metrics to evaluate the model is currently not feasible.

In the early stages of research on spatializing ground temperatures in high-altitude permafrost regions, researchers often used three-dimensional zonality based on longitude, latitude, and elevation for spatial prediction (Nan et al., 2013). While this approach may introduce systematic errors, particularly when predicting extreme high

and low values, it effectively captures the overall spatial trends of ground temperature on the QTP.

Here, we also employed multiple linear regression, incorporating longitude, latitude, and elevation as variables, to establish the spatial distribution of $MAGT_{15m}$. For ease of comparison, this prediction is referred to as LLE in the following discussion. The LLE result was used as the ground truth assumption to evaluate the performance of the RF method and other approaches.

Figure R3 illustrates the percentage frequency distribution of $MAGT_{15m}$ predicted by different methods. The comparison shows that the histograms of the support vector regression (SVR) and generalized additive model (GAM) methods closely resemble that of LLE method, exhibiting a high degree of consistency (Figure R3 a). In contrast, the histograms of the RF and XGBoost methods deviate significantly from LLE. The values predicted by RF and XGBoost are mainly concentrated between -3 °C and 1 °C, with notable peaks around 0 °C and 2 °C, where their frequency is substantially higher than in the LLE distribution. It is evident that the predictions from the SVR and GAM methods align well with the broader spatial temperature pattern of the QTP, as represented by LLE method. On the other hand, methods such as the RF and XGBoost fail to accurately capture this overarching spatial trend.

[Figure]

***Figure R3:*** *Percentage frequency distribution of predicted $MAGT_{15m}$ across the Qinghai-Tibet*

*Plateau permafrost regions with different methods (LLE-three dimensional zonality method based on longitude, latitude and elevation; SVR-support vector regression, GAM-generalized additive model, RF-random forest and XGBoost- extreme gradient boosting).*

In addition to comparing the distribution patterns of the histograms, we used the LLE result as the ground truth to assess the accuracy of the four methods (Table R5). The SVR method exhibited the lowest errors, with ME, MAE, and RMSE values of 1.36, 1.60, and 2.60 °C, respectively, along with the highest $R^2$ value of 0.46. The GAM method showed slightly higher errors than SVR, resulting in a slightly lower $R^2$ value of 0.39. In contrast, the $R^2$ values for RF and XGBoost were only -0.05 and 0.03, respectively, accompanied by relatively larger errors. This suggests that the spatial predictions from these methods are only weakly related to the spatial variation trend observed in the LLE method.

**Table R5:** *Accuracy evaluation of the four models compared to the result of LLE method[\*].*

| Performance | SVR | GAM | RF | XGBoost |
|---|---|---|---|---|
| $R^2$ | 0.46 | 0.39 | -0.05 | 0.03 |
| ME (°C) | 1.36 (±2.21) | 1.37 (±2.41) | 1.99 (±3.04) | 1.76 (±3.00) |
| MAE (°C) | 1.60 (±2.04) | 1.67 (±2.21) | 2.32 (±2.79) | 2.23 (±2.67) |
| RMSE (°C) | 2.60 (±6.53) | 2.77 (±8.77) | 3.63 (±8.19) | 3.47 (±8.07) |

[\*]*SVR, support vector regression; GAM, generalized additive model; RF, random forest; XGBoost, extreme gradient boosting. $R^2$, ME, MAE and RMSE with 1 standard deviation.*

Although the LLE method has been used as a reference for comparing the performance of different machine learning models, it is important to note that the evaluation remains largely qualitative. This is because the LLE result is also empirical in nature, commonly used in earlier studies with limited data availability. As such, it does not represent a strictly quantitative assessment of accuracy. Nevertheless, this comparison offers a valuable new perspective for evaluating the performance of different models. Ultimately, the comparisons conducted above clearly indicate that the

RF model, when applied to the current ground temperature dataset, is not optimally suited for predicting permafrost temperatures in the QTP.

The initial selection of the SVR method in this study was primarily informed by prior advanced research. For instance, Ran et al. (2021) performed a comparative analysis of various methods for ground temperature prediction on the QTP, highlighting the superior performance of SVR. Given the overlap of study area and the higher accuracy of SVR within the context of this study, we were further motivated to adopt this approach. Moreover, SVR provides deterministic results, which makes it especially well-suited for tasks requiring stability and reproducibility, ensuring transparency and consistency in the findings. As observational data continues to accumulate, the deterministic nature of SVR enables a robust evaluation of how new data influences the resulting predictions.

In conclusion, the performance disparities among different predictive methods are clear and can largely be attributed to the inherent characteristics of each algorithm, particularly how each method extracts patterns from the data. Based on a comparative analysis of the results from various methods and informed by previous research, SVR remains the most suitable approach for this study. While other methods similar to RF, such as XGBoost and KNN, could be evaluated, providing an exhaustive comparison within the confines of a single manuscript is challenging. At this stage, further comparisons might risk compromising the coherence and focus of the paper, which is why no additional modifications have been made in the revised manuscript. Nevertheless, we are committed to publishing our responses to your feedback to ensure the transparency of our methodology and potentially offer further insights to the readership. Your comments have highlighted the critical steps in spatial ground temperature prediction, and we plan to conduct a more focused, in-depth analysis of the predictive performance discrepancies between different methods, particularly with respect to the limitations of RF when applied to the existing dataset.

To ensure the manuscript accurately reflects the changes, all modifications have been marked using the track changes feature.

We would like to once again express our sincere gratitude for your valuable feedback!

References:

Nan, Z., Huang, P., and Zhao, L.: Permafrost distribution modeling and depth estimation in the Western Qinghai-Tibet Plateau, Acta. Geogr. Sin., 68, 318-327, 2013 (in Chinese).

Ran, Y., Li, X., Cheng, G., Nan, Z., Che, J., Sheng, Y., Wu, Q., Jin, H., Luo, D., Tang, Z., and Wu, X.: Mapping the permafrost stability on the Tibetan Plateau for 2005–2015, Sci. China Earth Sci., 64, 62–79, doi:10.1007/s11430-020-9685-3, 2021.

Ran, Y., Li, X., Cheng, G., Che, J., Aalto, J., Karjalainen, O., Hjort, J., Luoto, M., Jin, H., Obu, J., Hori, M., Yu, Q., and Chang, X.: New high-resolution estimates of the permafrost thermal state and hydrothermal conditions over the Northern Hemisphere, Earth Syst. Sci. Data, 14, 865–884, doi:10.5194/essd-14-865-2022, 2022.

---

## Author Response (AR3)

**Response to Referee #4**

We sincerely appreciate your review and the valuable comments on our manuscript, "Permafrost temperature baseline at 15 meters depth in the Qinghai-Tibet Plateau (2010–2019)" (MS No.: essd-2024-114). We have carefully addressed all the points raised and have made the necessary revisions accordingly.

I find the dataset on MAGT at a depth of 15 m and the permafrost model highly valuable for the permafrost research community. I recommend the publication of this paper after addressing the following minor revisions:

• The percentage of data measured in boreholes versus data estimated through linear regression used in permafrost modeling is unclear. It would be helpful to explicitly state this percentage in the Methodology section.

**Response:**

To clarify, the percentage of data estimated through linear regression has been explicitly stated in the Methodology section. The revised text now reads (Line 93-94):

*"As a result, approximately 1732 MAGT$_{15m}$ values were estimated through linear regression, which accounts for 75% of the total dataset."*

• Line 126: Please specify the resolution of the digital model used.

**Response:**

The resolution of the digital model has been specified in the manuscript as follows (Line 126):

*"The elevation **at 1 km resolution** was obtained from a dataset compiled by Amatulli et al. (2018)."*

• One interesting finding is the relationship between MAGT15m and slope. How do you explain the presence of colder permafrost in steep terrain? Additionally, could you provide the number of boreholes categorized within each slope class?

**Response:**

The Qinghai-Tibet Plateau (QTP) is broadly regarded as a mountain permafrost region. Although the extensive distribution of high-plateau has modified the spatial pattern of $MAGT_{15m}$ values to some extent, mountain permafrost is predominantly influenced by the elevation effect. Higher altitudes correspond to lower air temperatures, which subsequently result in lower permafrost temperatures. Our study also indicates that, in higher altitude areas, the decline in $MAGT_{15m}$ with increasing elevation is more pronounced compared to lower altitude regions.

Above the high-plateau, significant mountain ranges such as the Tangula, Kunlun Mountains, and Himalayas are located. The steep terrain typically associated with these mountains is the crucial factor contributing to the extremely low $MAGT_{15m}$ values observed in such areas. In addition to the higher elevations, these steep regions are less favorable to snow accumulation, which plays a critical role in maintaining lower permafrost temperatures. Furthermore, steep slopes are often characterized by exposed bedrock or scree, materials with higher thermal conductivity, which facilitates the transfer of cold energy to the subsurface during the winter season, thereby intensifying the cooling of permafrost. Thus, at the plateau scale, steep regions are concentrated at higher elevations, and their unique snow accumulation and surface cover characteristics further intensified the decline in $MAGT_{15m}$.

Regarding the number of boreholes categorized within the slope classes, the data is as follows: 133 boreholes in flat terrain (slope < 2°), 76 in gentle terrain (2° to 8°), 19 in moderate terrain (8° to 17°), and 3 in steep terrain (slope > 17°). Statistically, 90% of the boreholes in this study are in flat and gentle slope terrain. This distribution is largely attributable to the harsh climatic conditions of the QTP, the poor accessibility of road infrastructure, and the inherent difficulties associated with conducting drilling operations in steep terrain.

• The classification of slope classes may be unclear from a geomorphological perspective. To enhance clarity, please include a reference supporting the chosen slope

class intervals.

**Response:**

To address your concern regarding the geomorphological perspective of the slope classification, we have clarified the classification method in the revised manuscript. Specifically, we adopted the slope class intervals from the USDA Soil Survey Manual, and this reference has been added to both the Results and Reference sections as follows (Lines 193 and 381-382):

*"Considering the slope distribution pattern within the study areas, we aggregated the slope gradients into four classes: flat (slope < 2°), gentle (2° to 8°), moderate (8° to 17°), and steep (> 17°)* **(Soil Science Division Staff, 2017)***."*

*Reference: "Soil Science Division Staff: Soil survey manual, Ditzler C., Scheffe K., and Monger H.C. (eds.), USDA Handbook 18, Government Printing Office, Washington, D.C., 2017."*